# Diabetes impairs wound healing by Dnmt1-dependent dysregulation of hematopoietic stem cells differentiation towards macrophages

Jinglian Yan[1], Guodong Tie[1], Shouying Wang[1], Amanda Tutto[1], Natale DeMarco[1], Lyne Khair[1], Thomas G. Fazzio[2] & Louis M. Messina[1]

People with type 2 diabetes mellitus (T2DM) have a 25-fold higher risk of limb loss than non-diabetics due in large part to impaired wound healing. Here, we show that the impaired wound healing phenotype found in T2D mice is recapitulated in lethally irradiated wild type recipients, whose hematopoiesis is reconstituted with hematopoietic stem cells (HSCs) from T2D mice, indicating an HSC-autonomous mechanism. This impaired wound healing phenotype of T2D mice is due to a Nox-2-dependent increase in HSC oxidant stress that decreases microRNA let-7d-3p, which, in turn, directly upregulates Dnmt1, leading to the hypermethylation of *Notch1*, *PU.1*, and *Klf4*. This HSC-autonomous mechanism reduces the number of wound macrophages and skews their polarization towards M1 macrophages. These findings reveal a novel inflammatory mechanism by which a metabolic disorder induces an epigenetic mechanism in HSCs, which predetermines the gene expression of terminally differentiated inflammatory cells that controls their number and function.

---

[1] Diabetes Center of Excellence and Division of Vascular and Endovascular Surgery, University of Massachusetts Medical School, Worcester, MA 01655, USA. [2] Department of Molecular, Cell, and Cancer Biology, University of Massachusetts Medical School, Worcester, MA 01655, USA. Jinglian Yan and Guodong Tie contributed equally to this work. Correspondence and requests for materials should be addressed to L.M.M. (email: Louis.Messina@umassmemorial.org)

About 350 million people worldwide have type 2 diabetes mellitus (T2DM) and this is expected to grow to 440 million by 2030. One of the great scourges of diabetes is the threat of limb loss[1]. Of the one million people who undergo leg amputation annually worldwide, 75% are performed on people who have T2DM[2]. Impaired wound healing is a hallmark of the pathophysiology of diabetic foot ulcers and limb amputations. Despite the magnitude of these clinical consequences, the mechanism by which T2DM impairs wound healing remains unknown.

Normal wound healing involves three overlapping phases: (1) acute inflammation, (2) angiogenesis/proliferation, and (3) remodeling. Monocytes/macrophages derived from hematopoietic stem cells (HSCs) represent the most abundant inflammatory cell types during all stages of wound healing[3]. Monocyte-derived macrophages can polarize into either M1 (classically activated) or M2 (alternatively activated) subtypes in the presence of specific cues when recruited into peripheral tissues and wounds[4,5]. In general, M1 macrophages are characterized by their pro-inflammatory function. M2 macrophages mainly induce angiogenesis and tissue remodeling in the early phases of wound healing as well as reduce inflammation in the late phase of wound healing. T2DM wounds are characterized by excessive and prolonged inflammation due to the predominant presence of M1 macrophages[6–9]. However, the mechanism that causes this excessive and prolonged inflammation remains unknown[10–12].

Epigenetic modifications, including DNA methylation, histone modification, and small non-coding RNAs, are essential in maintaining proper lineage commitment and self-renewal of HSCs[13–16]. A recent study suggested that T2DM decreases the repressive histone methylation marker H3K27me3 in the promoter of the IL-12 gene in bone marrow progenitors. This epigenetic signature is passed down to terminally differentiated wound-resident macrophages in T2DM mice[8]. Furthermore, we have recently shown that the mechanism by which hypercholesterolemia increases cancer incidence involved an HSC-autonomous mechanism. In this circumstance, hypercholesterolemia-induced HSC oxidant stress increased the expression of miRNA101c that downregulated Tet1 expression in HSCs, which in turn reduced the number and function of terminally differentiated innate immune cells[17]. Thus, these changes in terminally differentiated cells are actually predetermined at the level of HSCs.

Taken together, these findings lead to a potentially novel paradigm that T2DM may epigenetically "preprogram" HSCs to reduce their differentiation towards monocytes and increase their polarization towards M1 macrophages and thereby negatively impact wound repair. However, there are two critical links missing in this hypothesis. First, even under conditions of normal wound healing, the evidence for a mechanism(s) that regulates monocyte tissue infiltration and the dynamics of their M1/M2 polarization is scarce. Second, the mechanisms by which T2DM reduces monocyte/macrophage tissue infiltration and their M1/M2 polarization remain unknown.

This study hypothesizes that T2D impairs wound healing by inducing oxidant stress-dependent epigenetic modifications in HSCs that reduce HSC differentiation towards monocytes/macrophages and disrupt the balance in M1/M2 polarization during the three phases of wound healing. In order to test our hypothesis, we generated a chimeric mouse model in which hematopoiesis was reconstituted in lethally irradiated WT recipient mice with HSCs from either db/db or WT mice. Here, we show for the first time that T2DM induces an HSC-autonomous mechanism that causes impaired wound healing. Specifically, T2DM causes a Nox-2-induced oxidant stress in HSCs that decreased microRNA let-7d-3p, which, in turn, directly increased the expression of DNA methyltransferase 1 (Dnmt1), a key enzyme mediating DNA methylation. Dnmt1-dependent repressive modifications reduced the expression of Notch1, PU.1, and Klf4 genes that are central in the differentiation of HSCs towards monocytes/macrophages. From a larger perspective, these novel findings reveal a new mechanism that regulates inflammation: T2DM induces changes in gene expression in HSCs that reduces the number and function of terminally differentiated inflammatory cells.

## Results

**T2DM reduces the differentiation of HSCs towards macrophages.** Diabetes impairs wound healing thereby making patients with T2DM susceptible to chronic non-healing wounds that often culminate in limb amputations[18]. We found that the wound closure rates in T2D mice (db/db mice or WT mice fed high fat diet (HFD)) were significantly slower than those in WT mice (Fig. 1a, b; Supplementary Fig. 1a). Histological analysis revealed a longer distance between epithelial tips and a longer distance between the edges of the panniculus carnosus in T2D wound tissues at days 3, 7, and 14 after wound induction (Fig. 1c, d, f), suggesting that the re-epithelialization and wound contraction were significantly impaired in T2D mice. Furthermore, the wound tissues from T2D mice showed much less granulation tissue, resulting in a thinner and more fragile epithelium (Fig. 1e, f). Revascularization was also reduced in the wounds of T2D mice, as measured by artery and total vessel density (Fig. 1g, h, i). These results indicate that wound healing kinetics are significantly impaired in T2D mice.

Monocytes and macrophages are the major cellular components that promote wound healing. Both the proportion and absolute number of $CD115^+CD11b^+$ monocytes were significantly reduced in the bone marrow of db/db and HFD mice (Fig. 2a, b, c; Supplementary Fig. 1b). We also measured other terminally differentiated blood cells derived from hematopoietic cells. With the exception of an increase in red blood cells, T cells, B cells, and granulocytes numbers were not significantly different in db/db mice (Supplementary Fig. 1c). Following the induction of cutaneous wounds, total macrophage infiltration in db/db mice was significantly lower on day 3 (inflammatory phase) and day 7 (new tissue formation phase), but significantly greater on day 14 (tissue remodeling phase) than in WT mice (Fig. 2d, e; Supplementary Fig. 2).

In addition, we compared the dynamic changes in M1/M2 polarization during the three phases of wound healing in WT and db/db mice. Both M1 and M2 macrophages were significantly reduced during the inflammatory phase in db/db mice (Fig. 2f). During the new tissue formation phase, M1 macrophages in the wounds of db/db mice were increased to a level much greater than those in the wounds of WT mice, while M2 macrophages remained at very low levels (Fig. 2g; Supplementary Figs. 2 and 3). In the remodeling phase, M1 macrophages were still at a significantly greater concentration in the wounds of db/db mice, while M2 macrophages were at a level equivalent to the wounds of WT mice (Fig. 2h).

In order to test the hypothesis that T2DM causes an HSC-autonomous defect that impairs wound closure kinetics by dysregulation of HSC lineage specification towards monocytes and macrophage polarization, we investigated wound closure kinetics in a chimeric model whereby hematopoiesis was reconstituted in lethally irradiated WT recipient mice with HSCs from either db/db or WT mice. The wound closure rates in WT mice were significantly delayed in the recipients reconstituted with db/db HSCs (Fig. 3a, b). We also observed a significant decrease in the rates of re-epithelialization (Fig. 3c, f; Supplementary Fig. 4a), wound contraction (Fig. 3d, f; Supplementary Fig. 4a) and revascularization (Supplementary Fig. 4b, c, d), as

well as a significant decrease in granulation tissue (Fig. 3e, f; Supplementary Fig. 4a). The recipients reconstituted with *db/db* HSCs also recapitulated the defects in monocyte and macrophage

frequency (Fig. 3g, h; Supplementary Fig. 5a, b) as well as the skewed M1 polarization of macrophages in wounds (Fig. 3i, j, k; Supplementary Fig. 5c, d). No change was found in T cells, B cells, granulocytes and red blood cells (Supplementary Fig. 4e). These results indicate that T2DM reduces the differentiation of HSCs towards monocytes/macrophages, and skews their polarization towards the M1 phenotype in an HSC-autonomous manner that results in a pro-inflammatory environment that impairs wound healing kinetics.

**T2DM induces Nox-2-dependent upregulation of Dnmt1 in HSCs**. In previous studies, our group has shown that cardiovascular risk factors such as hypercholesterolemia, increases HSC oxidant stress[17]. Similarly, HSCs from T2D mice had significantly greater oxidant stress than HSCs from WT mice, as shown by the increased DCF-DA staining (Fig. 4a; Supplementary Fig. 6a). N-acetylcysteine (NAC) supplementation reversed the oxidant stress and restored the monocyte concentration in the bone marrow of *db/db* mice (Fig. 4a, b).

The Nox family of NADPH oxidases is an important source of reactive oxygen species[19]. Among them, Nox-1, Nox-2, and Nox-4 have been reported to be expressed in HSCs[20,21]. We observed significantly increased Nox-2 and Nox-4 expression in HSCs from *db/db* mice (Fig. 4c; Supplementary Fig. 6b, c). The inhibition of Nox-2 and Nox-4 reduced the oxidant stress in HSCs from *db/db* mice (Fig. 4d, e). Since Nox-2 induced a much higher level of oxidant stress in HSCs from *db/db* and HFD mice (Fig. 4d, e; Supplementary Fig. 6b), we focused on identifying the role of Nox-2 in HSC differentiation towards monocytes/ macrophages in *db/db* mice.

DNA methylation plays an important role in HSC commitment to lymphoid and myeloid lineages. We have recently shown that hypercholesterolemia causes HSC oxidant stress, which in turn induces an epigenetic mechanism that dysregulated HSC lineage commitment[17]. DNA methylation of CpG nucleotides, a key epigenetic modification, is established and maintained by a family of DNA methyltransferases (Dnmts): Dnmt1, Dnmt3a, and Dnmt3b[22]. We found that Dnmt1 was consistently upregulated in HSCs from both *db/db* and HFD mice, while Dnmt3b was only increased in HSCs from *db/db* mice (Fig. 4f; Supplementary Fig. 7a). Dnmt1 mRNA and protein levels were also increased in bone marrow monocytes and wound macrophages (Supplementary Fig. 7b, c, d). The inhibition of Nox-2 or the treatment with NAC both restored the expression of Dnmt1 in *db/db* HSCs to a level comparable to that in WT HSCs, but did not change the mRNA or the protein levels of Dnmt3a and Dnmt3b (Fig. 4g, h). Consistent with the upregulation of Dnmt1, the total DNA methylation status in *db/db* HSCs was higher than in WT HSCs (Supplementary Fig. 7e). These results indicate that T2DM upregulates the expression of Dnmt1 and Dnmt3b. However, only Dnmt1 was upregulated by Nox-2-dependent oxidant stress.

**Let-7d-3p mediates the upregulation of Dnmt1 in T2D HSCs**. In order to identify the mechanism regulating Dnmt1 expression in T2D HSCs, we next investigated the role of microRNAs, which are non-coding RNAs that regulate gene expression by binding to the 3′ untranslated region (UTR) of protein coding genes[23]. To identify microRNAs that are regulated by T2DM, we performed a microRNA microarray analysis of HSCs from WT and *db/db* mice. Among the differentially expressed microRNAs, Let-7d-3p, which is predicted to directly target Dnmt1, showed a significant oxidant stress-dependent downregulation in HSCs from *db/db* mice (Fig. 5a; Supplementary Fig. 8). To determine if let-7d-3p binds directly to Dnmt1, we cloned the mouse Dnmt1 3′ UTR

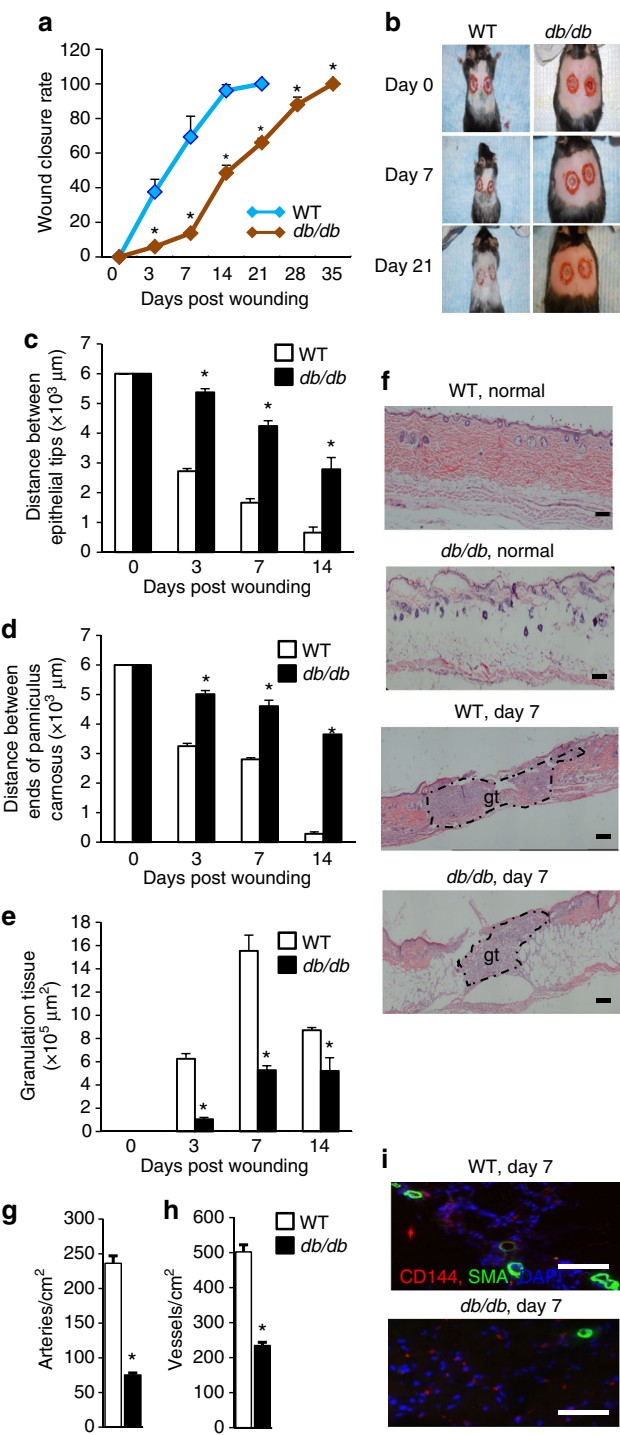

**Fig. 1** Morphometric analysis of impaired wound healing in Type 2 diabetic mice. **a** Wound closure rate measurement ($n = 8$, *$p < 0.05$ vs. WT). **b** Representative wound images. **c–e** Histological quantification of distance between epithelial tips/ends of panniculus carnosus and granulation tissue ($n = 4$, *$p < 0.05$ vs. WT). **f** Representative H&E staining wound images, magnification ×40. Scale bar, 100 μm. gt, granulation tissue. **g, h** Immunohistochemical (IHC) quantification of vascularization by CD144 and α-SMA staining. **i** Representative IHC staining images, magnification ×200. Scale bar, 100 μm. Results are expressed as means ± SEM. Two-tailed unpaired Student's *t* test was used for **a**, **c–e**, **g**

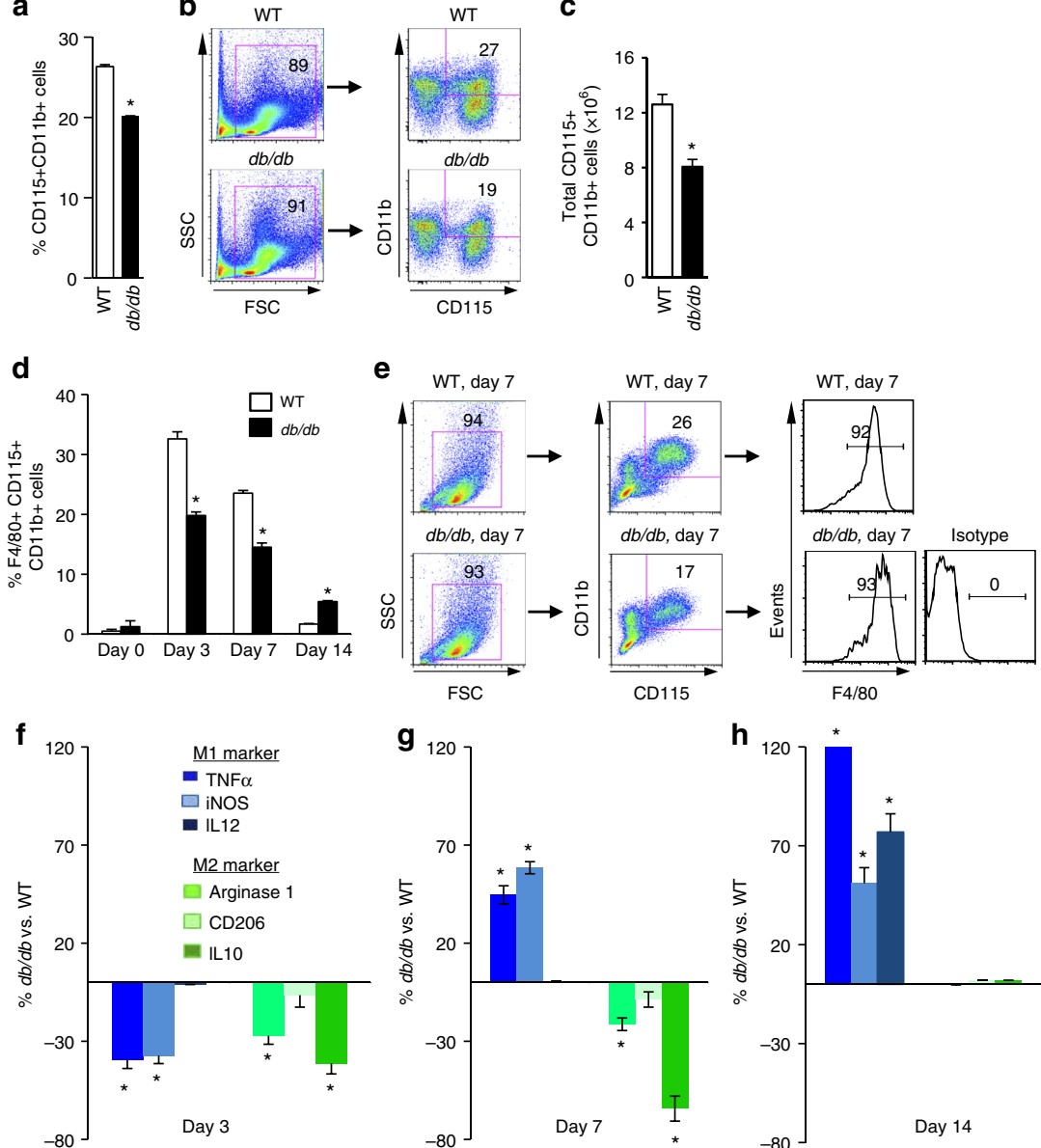

**Fig. 2** Type 2 diabetes reduces macrophage infiltration into wounds. **a** Quantification of monocytes concentration in bone marrow by flow cytometry ($n = 6$, $*p < 0.05$ vs. WT). **b** Schematic of flow cytometry gating. **c** Quantification of absolute number of monocytes in bone marrow ($n = 6$, $*p < 0.05$ vs. WT). **d** Quantification of macrophage concentration in the cutaneous wounds by flow cytometry ($n = 6$, $*p < 0.05$ vs. WT). **e** Schematic of flow cytometry gating. **f–h** Quantification of M1/M2 polarization in the cutaneous wounds by flow cytometry ($n = 6$, $*p < 0.05$ vs. WT). Results are expressed as means ± SEM. Two-tailed unpaired Student's $t$ test was used for **a**, **c**, **d**, **f**, **g**, **h**

region (labeled as Dnmt1 in Fig. 5b) and determined the luciferase activity in HEK293T cells transfected with let-7d-3p mimic. Let-7d-3p significantly repressed the luciferase activity when we used an intact 3′ UTR construct, but did not affect the luciferase activity when the binding site of let-7d-3p in Dnmt1 3′ UTR region was mutated (labeled as Dnmt1 M1 and Dnmt1 M2) (Fig. 5b). In addition, a let-7d-3p inhibitor increased Dnmt1 expression in WT HSCs, while let-7d-3p mimic decreased Dnmt1 expression in *db/db* HSCs (Fig. 5c, d, e, f). Taken together, these results provide compelling evidence that the reduced level of let-7d-3p directly mediates the increased expression of Dnmt1 in HSCs.

**Upregulation of Dnmt1 in T2D HSCs impairs wound healing**. We next sought to analyze the role of Dnmt1 in HSC differentiation towards monocytes. The expression of Dnmt1 in *db/db*

HSCs was downregulated following the addition of an shRNA against Dnmt1 (Fig. 6a; Supplementary Fig. 9a, b). Under in vitro conditions, the knockdown of Dnmt1 in *db/db* HSCs consistently increased their differentiation towards monocytes/ macrophages (Supplementary Fig. 9c, d). WT recipients reconstituted with Dnmt1-knockdown *db/db* HSCs showed a significantly higher rate of wound healing than WT recipients reconstituted with *db/db* HSCs (Fig. 6b, c, d, e). In WT mice reconstituted with Dnmt1-knockdown *db/db* HSCs, the infiltration of monocytes/macrophages, as well as their M1/M2 polarization throughout the three phases of wound healing, were restored to levels comparable to those in WT mice reconstituted with WT HSCs (Fig. 6f, g, h, i). These findings indicate that sustained upregulation of Dnmt1 expression in HSCs mediates the deleterious effects of T2DM on wound healing by inhibiting their differentiation towards monocytes/macrophages and skewing them toward M1 polarization.

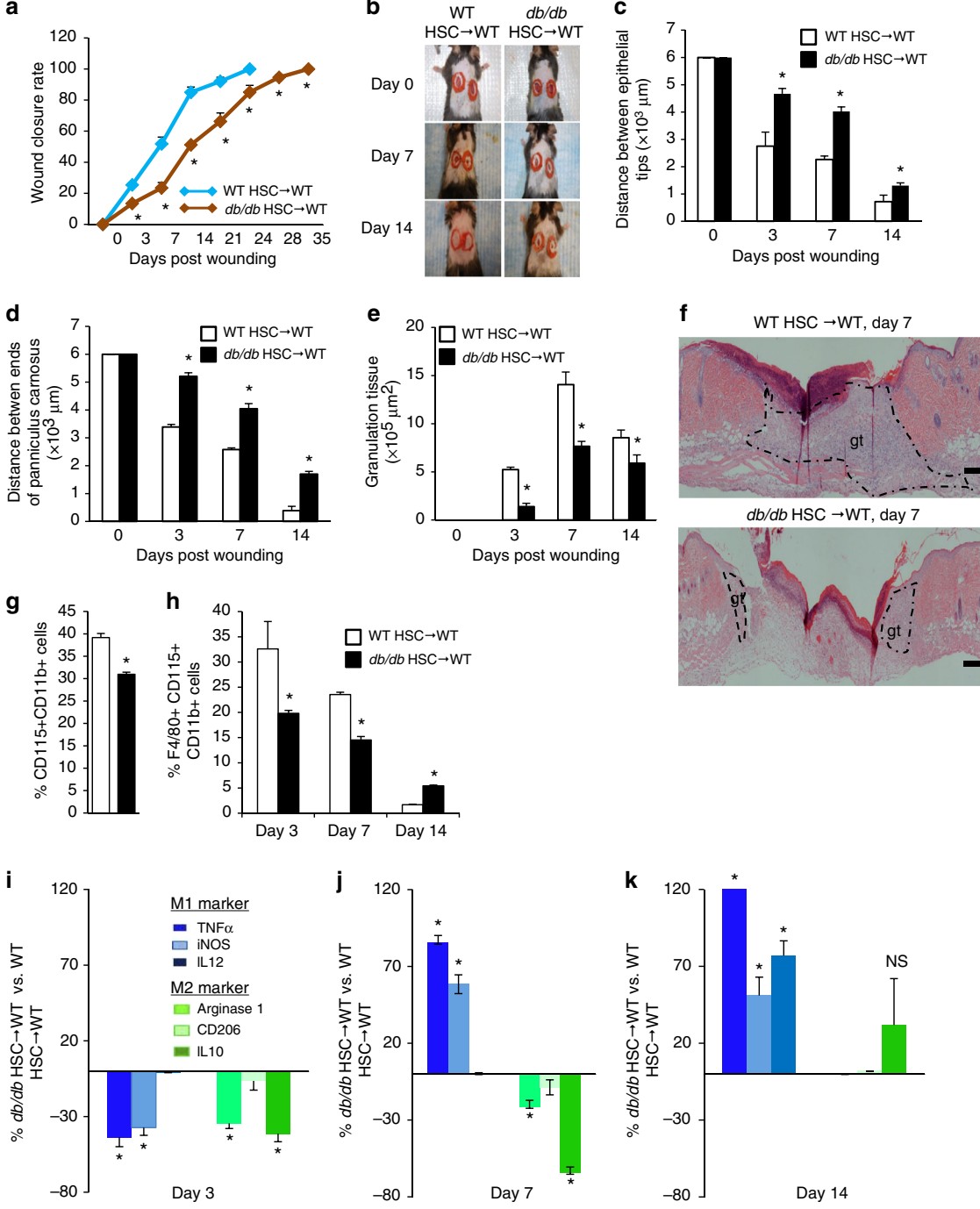

**Fig. 3** Type 2 diabetes impairs wound healing and monocytes/macrophages infiltration. **a**, **b** Wound closure rate measurement and representative images (n = 8, *p < 0.05 vs. WT HSC → WT). **c–e** Histological quantification of distance between epithelial tips/ends of panniculus carnosus and granulation tissue (n = 4, *p < 0.05 vs. WT HSC→WT). **f** Representative H&E staining wound images, magnification ×40. Scale bar, 100 μm. gt, granulation tissue. **g** Quantification of monocytes concentration in bone marrow by flow cytometry (n = 6, *p < 0.05 vs. WT HSC → WT). **h** Quantification of macrophage concentration in the cutaneous wounds by flow cytometry (n = 6, *p < 0.05 vs. WT HSC → WT). **i–k** Quantification of M1/M2 polarization in cutaneous wounds by flow cytometry. (n = 6, *p < 0.05 vs. WT HSC → WT). Results are expressed as means ± SEM. Two-tailed unpaired Student's t test was used for **a**, **c–e**, **g–k**

**Dnmt1 knockdown in HSCs increases wound healing in T2D mice**. In order to determine in vivo if Dnmt1-knockdown HSCs from T2D mice rescue normal wound healing, we transplanted Dnmt1-knockdown db/db HSCs into lethally irradiated T2D recipients, and then induced wounds 2 months after HSC transplantation. We found that the wound closure rate was significantly increased in T2D recipients transplanted with either

WT HSCs or Dnmt1-knockdown db/db HSCs. Meanwhile, the most impaired wound closure rate was observed in db/db recipients transplanted with db/db HSCs (Supplementary Fig. 10a, b). Histological analysis also showed that the transplantation of Dnmt1-knockdown db/db HSCs significantly improved re-epithelialization (Supplementary Fig. 11a, d), wound contraction (Supplementary Fig. 11b, d) and revascularization

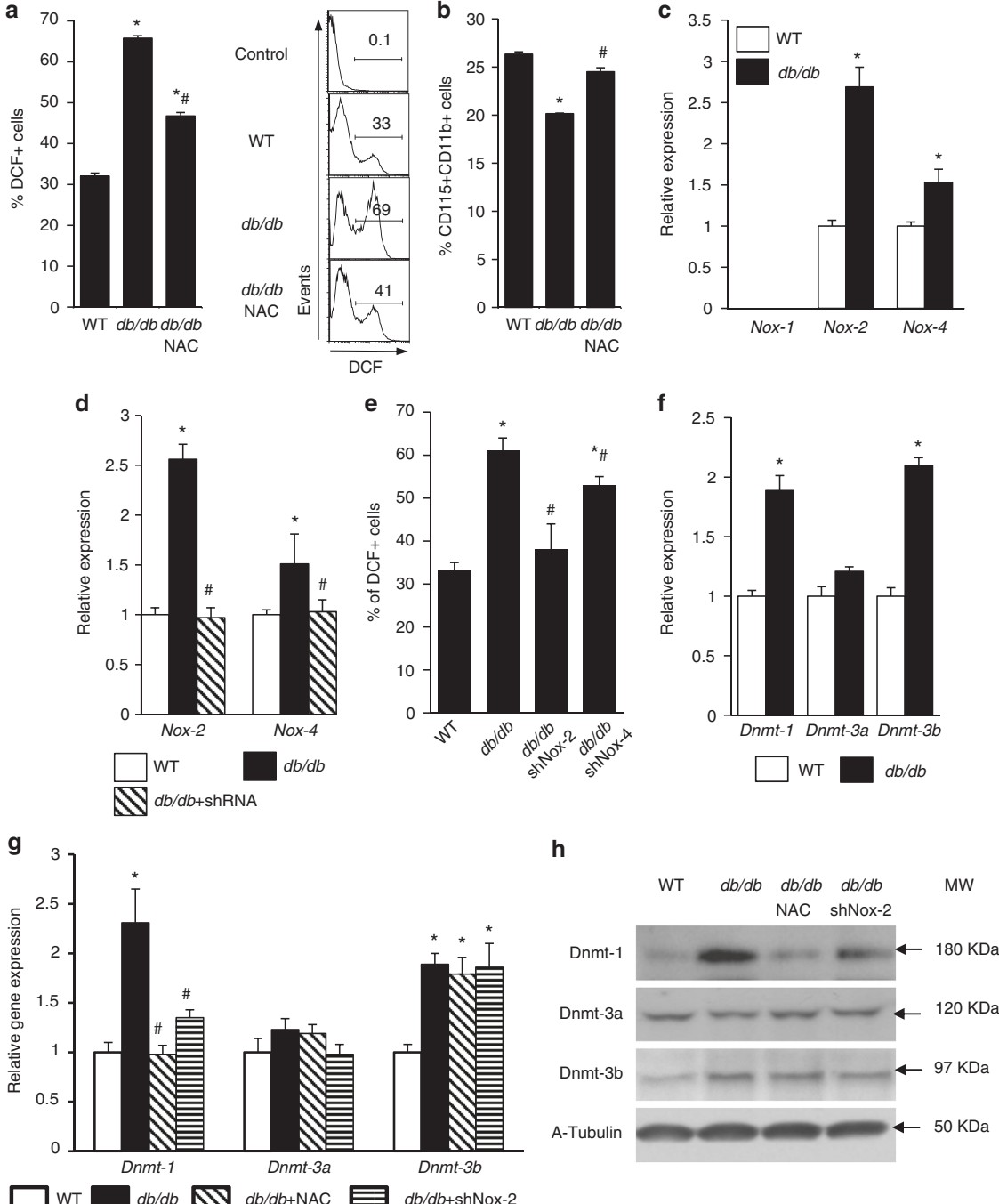

**Fig. 4** Nox-2-dependent increase in Dnmt1 expression. Quantification by flow cytometry of **a** DCF positive cells ($n = 6$, *$p < 0.05$ vs. WT; #$p < 0.05$ vs. db/db). **b** Monocytes concentration in bone marrow ($n = 6$, *$p < 0.05$ vs. WT; #$p < 0.05$ vs. db/db). **c** NADPH oxidase gene expression ($n = 6$, *$p < 0.05$ vs. WT). **d** Nox-2 and Nox-4 gene expression after knockdown of Nox-2 or Nox-4 ($n = 6$, *$p < 0.05$ vs. WT; #$p < 0.05$ vs. db/db). **e** Quantification of DCF positive cells after knockdown of Nox-2 or Nox-4 ($n = 6$, *$p < 0.05$ vs. WT; #$p < 0.05$ vs. db/db). **f** Dnmts gene expression ($n = 6$, *$p < 0.05$ vs. WT). **g** Dnmts expression after knockdown of Nox-2 or NAC antioxidant treatment ($n = 6$, *$p < 0.05$ vs. WT; #$p < 0.05$ vs. db/db). **h** Dnmts protein levels after knockdown of Nox-2 or NAC antioxidant treatment ($n = 3$, *$p < 0.05$ vs. WT; #$p < 0.05$ vs. db/db). Results are expressed as means ± SEM. One way ANOVA was used for **a**, **b**, **d**, **e**, **g**. Two-tailed unpaired Student's $t$ test was used for **c**, **f**

(Supplementary Fig. 4f, g, h), as well as significantly increased granulation tissue (Supplementary Fig. 11c, d). These findings indicate that the transplantation of autologous Dnmt1-knockdown HSCs from T2D mice may provide a potential therapeutic option for T2DM patients.

**Hyperinsulinemia impairs the differentiation of human HSCs.** Insulin resistance and the consequent hyperinsulinemia is a key feature of T2DM. Hyperinsulinemia-induced oxidant stress impairs the function of stem and progenitor cells[24,25]. In in vitro experiments we found that hyperinsulinemia reduced human HSC differentiation towards macrophages (Supplementary Fig. 12a, b). In addition, hyperinsulinemia skewed human macrophages towards M1 polarization (Supplementary Fig. 12c). Hyperinsulinemia also increased the expression of NOX-2 and DNMT1 in human HSCs (Supplementary Fig. 12d). These

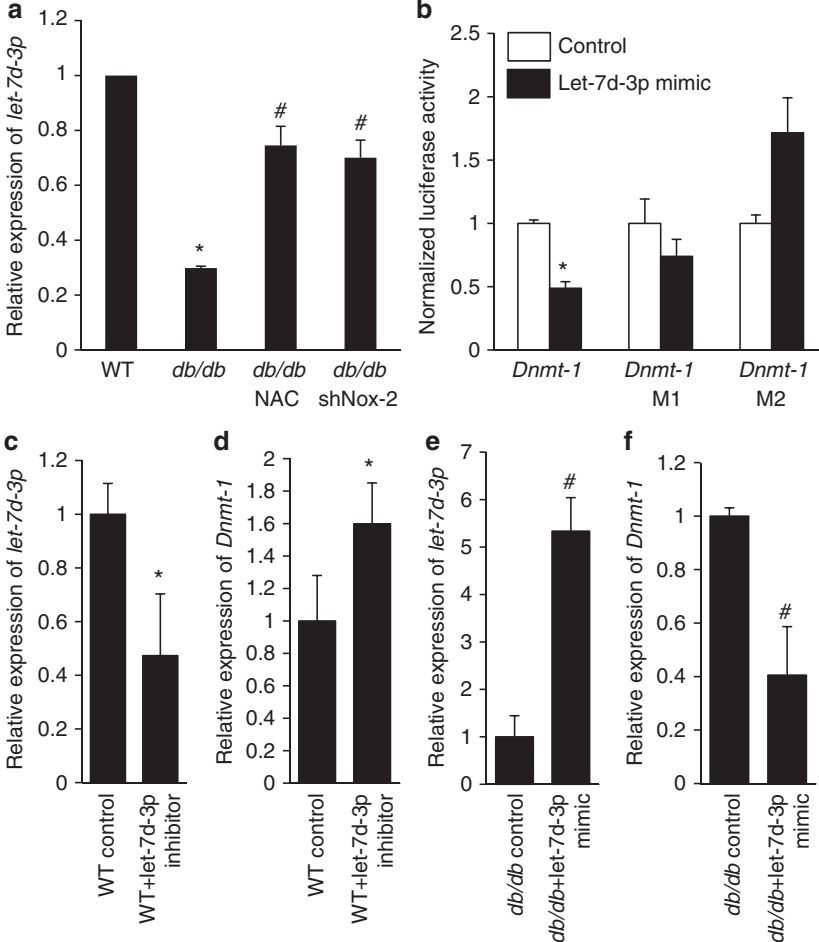

**Fig. 5** Dnmt1 is the direct downstream target of microRNA let-7d-3p. **a** Relative expression of *let-7d-3p* in *db/db* HSCs ($n = 6$, *$p < 0.05$ vs. WT; #$p < 0.05$ vs. *db/db*). **b** Luciferase activity assay ($n = 4$, *$p < 0.05$ vs. control). *Dnmt1* M1 represents mutant #1 and *Dnmt1* M2 represents mutant #2. **c** Relative expression of *let-7d-3p* in WT HSCs following transfection with a let-7d-3p inhibitor ($n = 6$, *$p < 0.05$ vs. WT control). **d** Relative expression of *Dnmt1* in WT HSCs following transfection of let-7d-3p inhibitor ($n = 6$, *$p < 0.05$ vs. WT control). **e** Relative expression of *let-7d-3p* in *db/db* HSCs following transfection with let-7d-3p mimic. **f** Relative expression of *Dnmt1* in *db/db* HSCs following transfection of let-7d-3p mimic ($n = 6$, #$p < 0.05$ vs. *db/db* control). Results are expressed as means ± SEM. One way ANOVA was used for **a**. Two-tailed unpaired Student's *t* test was used for **b**–**f**

findings suggest that hyperinsulinemia impairs the differentiation of human HSCs towards monocytes/macrophages by oxidant stress-dependent upregulation of DNMT1.

**Upregulated Dnmt1 in T2D HSCs represses *Klf4*, *PU.1*, and *Notch1*.** Transcription factors play a major role in the regulation of HSC lineage commitment. Among them, PU.1 and Klf4 were reported to guide monocyte and macrophage subtype-specific programs[26–31], while Notch1 was shown to be an important extrinsic regulator of myeloid commitment that targets the transcription factor PU.1[32–34]. We found that the expression of *Notch1*, *PU.1*, and *Klf4* in HSCs from T2D mice was significantly decreased in an oxidant stress-dependent manner (Fig. 7a). Furthermore, knocking down Dnmt1 increased the expression of *Notch1*, *PU.1*, and *Klf4* in *db/db* HSCs (Fig. 7b). These results suggest that oxidant stress-dependent upregulation of Dnmt1 in HSCs from T2D mice-induced repressive modifications of *Klf4*, *PU.1*, and *Notch 1*.

To ascertain the pathological significance caused by the decrease in Notch1, PU.1, and Klf4 on the differentiation and polarization of monocytes and macrophages, we knocked down the expression of all three genes in WT HSCs. Knockdown of Notch1, PU.1, and Klf4 reduced WT HSC differentiation towards monocytes/macrophages (Fig. 7c, d). These results indicate that

Dnmt1-dependent downregulation of *Notch1*, *PU.1*, and *Klf4* in HSCs is likely responsible for the attenuation of monocyte/macrophage differentiation in T2D mice, at least under in vitro conditions.

Knockdown of Klf4 in WT HSCs skewed monocyte polarization toward the M1 phenotype (Fig. 7e). This finding is consistent with a previous study that showed myeloid Klf4 deficiency increased pro-inflammatory M1 macrophages and facilitated the inflammatory response during wound healing[35,36]. These results support our findings that downregulation of *Klf4* in T2DM HSCs is most likely responsible for the polarization towards M1 macrophages. In addition, this supports the novel concept that changes in gene expression of progenitor and terminally differentiated inflammatory cells can be "preprogramed" at the level of HSCs.

**Dnmt1 induces DNA hypermethylation of *Notch1*, *PU.1*, and *Klf4*.** To test if the upregulation of Dnmt1 increases the repressive modification of the genes critical in the differentiation of HSCs towards monocytes/macrophages, we measured the DNA methylation status of *Notch1*, *PU.1*, and *Klf4* genes in HSCs from WT and *db/db* mice. Pyrosequencing analysis showed a significant increase in DNA methylation in *Notch1*, *PU.1*, and *Klf4* in *db/db* HSCs. On the other hand, increased expression of let-7d-

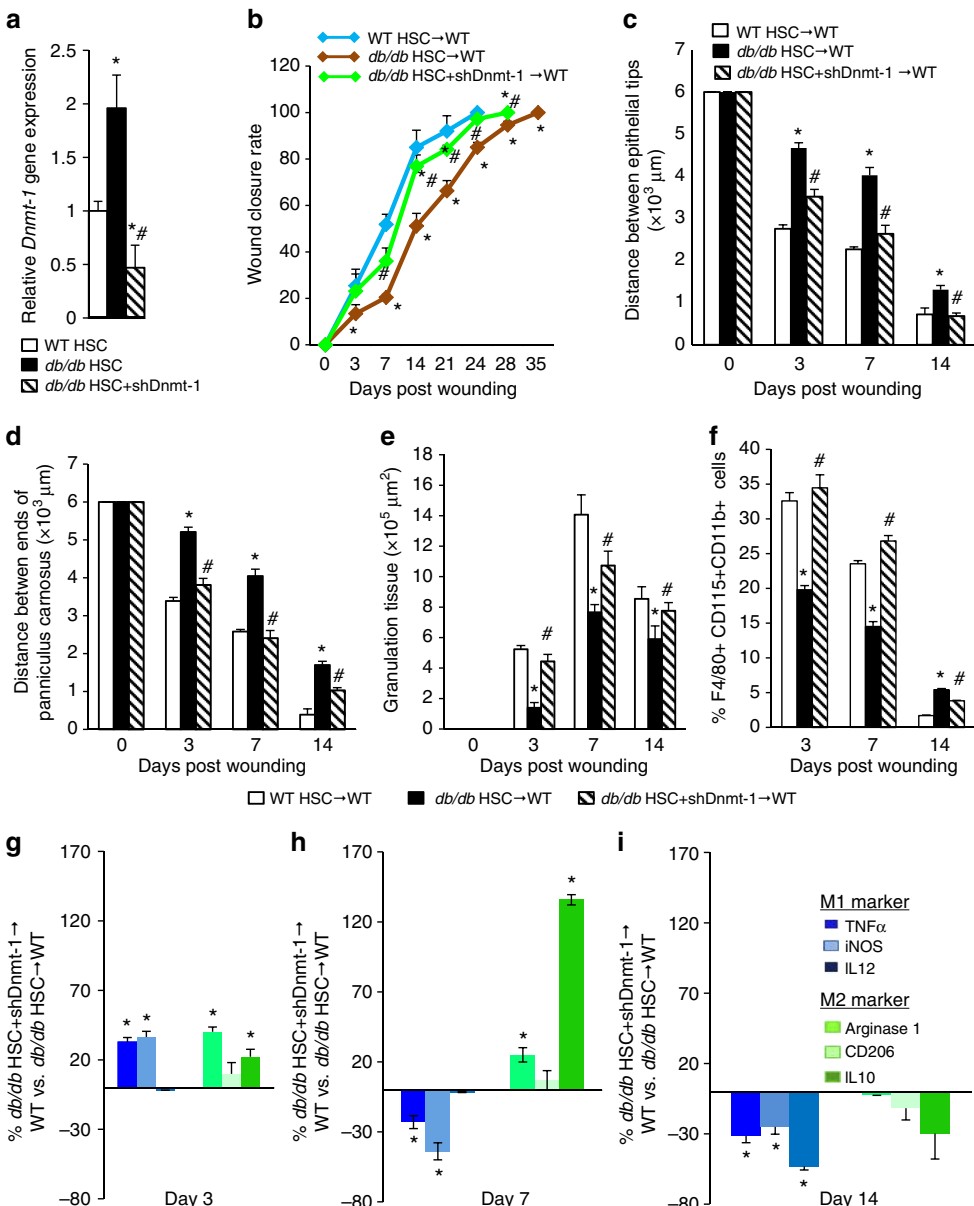

**Fig. 6** Knockdown of Dnmt1 in *db/db* HSCs increases wound closure rate. **a** qRT-PCR quantification of *Dnmt1* expression ($n = 6$, \*$p < 0.05$ vs. WT HSC, #$p < 0.05$ vs. *db/db* HSC). **b** Wound closure rate measurement ($n = 8$, \*$p < 0.05$ vs. WT HSC → WT, #$p < 0.05$ vs. *db/db* HSC → WT). **c–e** Histological quantification of distance between epithelial tips/ends of panniculus carnosus and granulation tissue ($n = 4$, \*\*$p < 0.05$ vs. W T HSC→ WT, #$p < 0.05$ vs. *db/db* HSC→ WT). **f** Quantification of macrophage concentration in the cutaneous wounds by flow cytometry ($n = 6$, \*$p < 0.05$ vs. WT HSC → WT, #$p < 0.05$ vs. *db/db* HSC → WT). **g–i** Quantification of M1/M2 polarization in cutaneous wounds by flow cytometry. ($n = 6$, #$p < 0.05$ vs. *db/db* HSC → WT). Results are expressed as means ± SEM. One way ANOVA was used for **a–f**. Two-tailed unpaired Student's *t* test was used for **g–i**

3p and knockdown of Dnmt1 in *db/db* HSCs decreased the methylation of *Notch1*, *PU.1*, and *Klf4* (Fig. 8a; Supplementary Figs. 13 and 14). These results suggest that Dnmt1-induced DNA methylation inhibits the expression of *Notch1*, *PU.1*, and *Klf4* in HSCs from *db/db* mice.

Dnmt1 has been shown to contribute to the regulation of histone modifications. Indeed, ablation of Dnmt1 reduces dimethylation (H3K9me2) and trimethylation (H3K9me3) at H3K9 and increases H3K9 acetylation (H3K9ac)[37,38]. Therefore, we performed ChIP-PCR to test if H3K9 methylation silences the expression of *Klf4*, *PU.1*, and *Notch 1*. Both the dimethylated (H3K9me2) and trimethylated (H3K9me3) histone proteins were increased in the promoters of *Klf4*, *PU.1*, and *Notch1* genes in HSCs from T2D mice (Fig. 8b; Supplementary Fig. 15a). However, no difference in H3K9Ac was recorded in the three

genes in WT or T2DM HSCs (Supplementary Fig. 15b). Both knockdown of Dnmt1 and increased expression of let-7d-3p in *db/db* HSCs decreased the histone methylation levels (Supplementary Figs. 15b; Fig. 16). These findings indicate that the oxidant stress-induced inhibition of let-7d-3p leads to an upregulation of Dnmt1 which in turn increases the repressive H3K9 methylation and decreases the expression of *Notch1*, *PU.1*, and *Klf4* in HSCs from T2D mice.

## Discussion

Traditionally, wound healing has been described as a process that requires a finely tuned orchestrated recruitment of terminally differentiated cells by different external cues to wound sites. Although it is well established that this process is delayed in T2DM, to date, the mechanism(s) responsible for such a delay

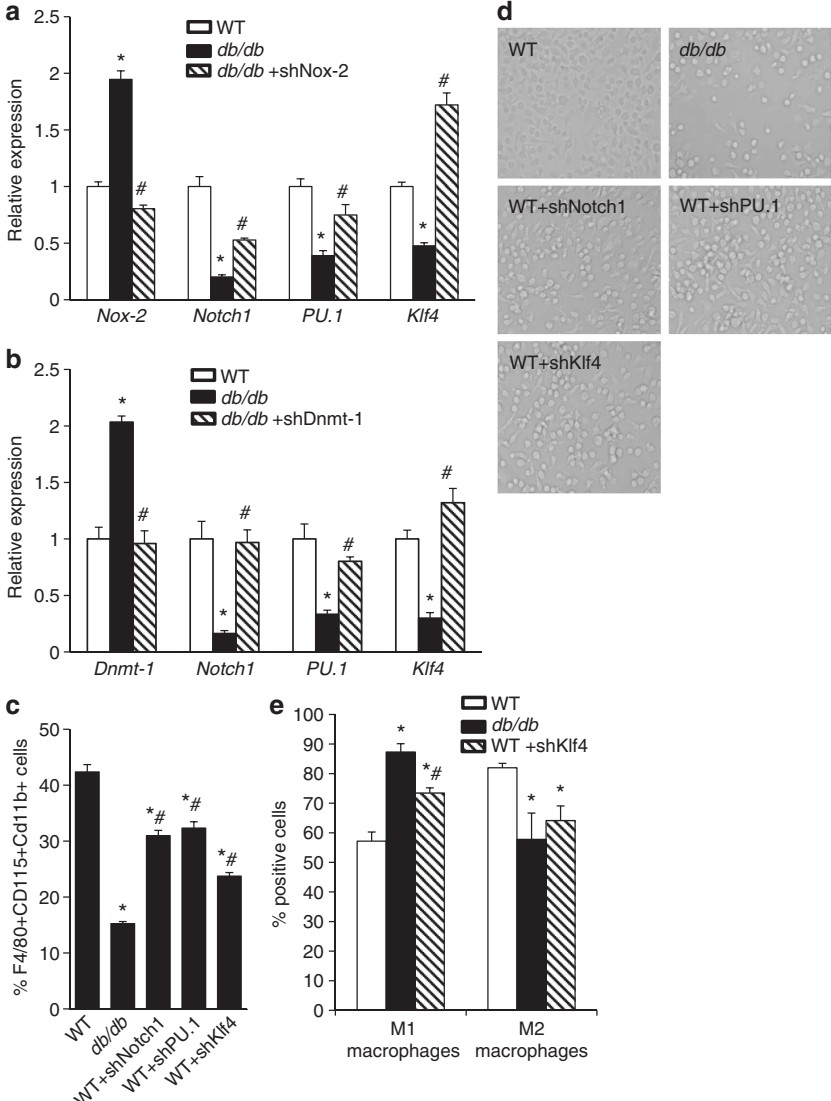

**Fig. 7** Oxidant stress-dependent downregulation of *Klf4*, *PU.1*, and *Notch1*. Gene expression analysis following. **a** Nox-2 knockdown ($n = 6$. *$p < 0.05$ vs. WT; #$p < 0.05$ vs. *db/db*). **b** Dnmt1 knockdown ($n = 6$. *$p < 0.05$ vs. WT; #$p < 0.05$ vs. *db/db*). **c** Flow cytometry analysis of HSC-induced differentiation towards macrophages under in vitro conditions ($n = 6$, *$p < 0.05$ vs. WT; #$p < 0.05$ vs. *db/db*). **d** Representative images. **e** Flow cytometry analysis of M1/M2 macrophages ($n = 6$. *$p < 0.05$ vs. WT; #$p < 0.05$ vs. *db/db*). Results are expressed as means ± SEM. One way ANOVA was used for **a**–**c**, **e**

had not been well defined. In this study, we show that T2DM impairs wound healing through an HSC-autonomous mechanism whereby a Nox-2-dependent increase in HSC oxidant stress decreases microRNA let-7d-3p, which, in turn, directly increases the expression of Dnmt1. This increase in Dnmt1 expression leads to the downregulation of the genes responsible for HSC differentiation towards monocytes/macrophages, and consequently reduces macrophage infiltration and skews the polarization towards M1 macrophages in the wounds of T2D mice.

One of the key findings of this study is that T2DM induces pathological HSC oxidant stress that can reduce the number and function of terminally differentiated inflammatory cells. In a recent study, Nox-2 has been shown to contribute to hyperinsulinemia-induced redox imbalance in endothelial cells[39]. Nox-2-induced oxidant stress has also been shown to play a critical role in insulin resistance-related endothelial cell dysfunction that eventually leads to vascular occlusive disease[40]. Our results show that oxidant stress-induced epigenetic mechanisms lead to changes in HSC gene expression that are carried down through progenitor cells to terminally differentiated cells. This

supports the concept that changes in gene expression of terminally differentiated inflammatory cells are actually epigenetically "preprogrammed" at the HSC level.

Thus, the concentration of macrophages in wounds of T2D mice is significantly lower than that in WT mice in early inflammatory and new tissue regeneration phases, but remains at significantly greater levels in the tissue remodeling phase. In agreement with previous findings, our study showed that M1 macrophages comprise the absolute majority of macrophages in the wounds of T2D mice[6–8].

Moreover, our study is the first to show that Dnmt1 in HSCs functions as a critical determinant in their differentiation towards monocytes and macrophages as well as in their polarization. Indeed, we show that Dnmt1 was specifically upregulated in HSCs from both *db/db* and HFD mice and this supports the fact that the impairment of wound healing is not due to other metabolic abnormalities of *db/db* mice. The inhibition of Dnmt1 in HSCs from T2D mice increased the differentiation from HSCs towards monocytes/macrophages and restored the balance of M1/M2 polarization

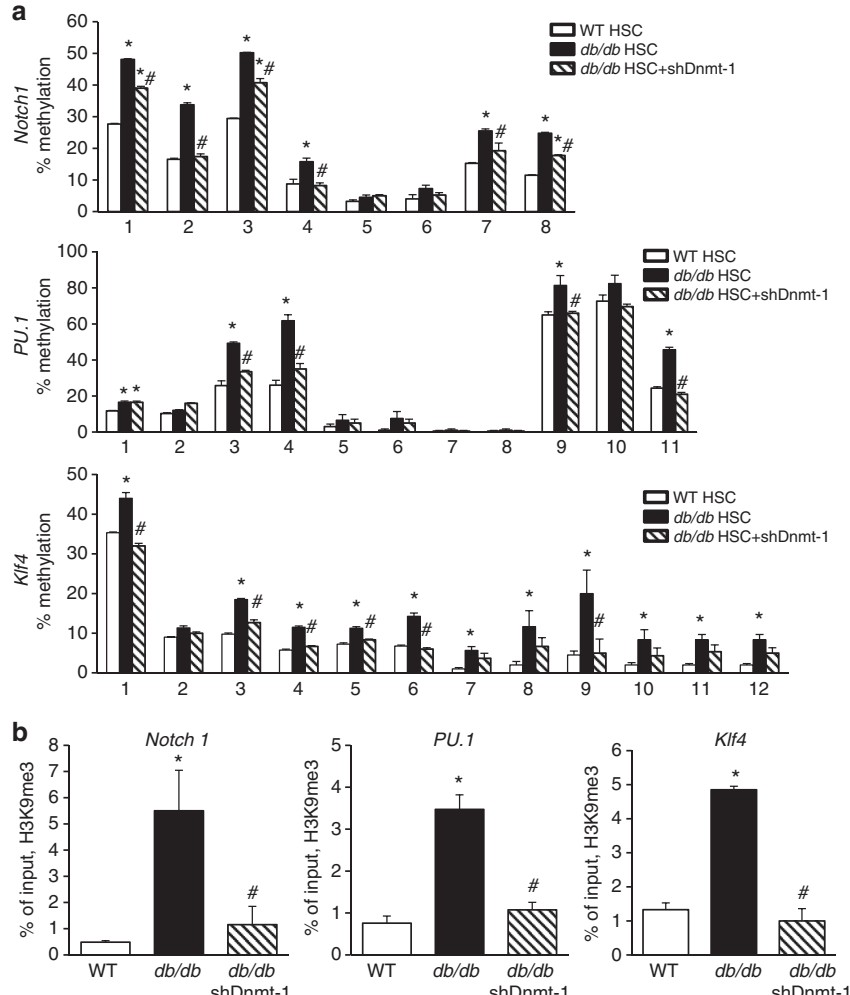

**Fig. 8** T2DM upregulates Dnmt1 in *db/db* HSCs. **a** DNA methylation analysis by pyrosequencing (*n* = 6, *$p$ < 0.05 vs. WT; #$p$ < 0.05 vs. *db/db*). **b** ChIP-PCR analysis of histone modifications (*n* = 6. *$p$ < 0.05 vs. WT; #$p$ < 0.05 vs. *db/db*). Results are expressed as means ± SEM. One way ANOVA was used for **a**, **b**

in vivo. More importantly, the transplantation with Dnmt1-knockdown HSCs from *db/db* mice restores the rate of wound healing in T2D recipients to levels similar to those seen in WT mice. These novel findings show that the dysregulated lineage priming of HSCs in T2D mice is the central underlying cause of impaired wound healing.

Notch1, PU.1, and Klf4 have been shown to be the most critical regulators of the differentiation and polarization of monocytes/macrophages. In our study, *Notch1*, *PU.1*, and *Klf4* were downregulated in HSCs from T2D mice. As is known, Notch1 deficiency leads to decreased macrophage recruitment and TNFα expression in response to wounding[34]. *Klf4* is a downstream target gene of *PU.1* and *Notch 1*[41,42] and induces an M2 phenotype while inhibiting the M1 phenotype. Klf4 deficiency in myeloid cells results in delayed wound healing and increased proinflammatory M1 macrophages[36]. In agreement with these findings, the T2DM-induced inhibition of *Notch1*, *PU.1,* or *Klf4* in HSCs reduced the differentiation and polarization of monocytes/macrophages under in vitro conditions.

The mechanism by which Dnmt1 regulates the expression of *Notch1*, *PU.1*, and *Klf4* is through the modification of their methylation status. Indeed, we found that the DNA methylation status of *Notch1*, *PU.1*, and *Klf4* increased in HSCs from *db/db* mice. Furthermore, Dnmt1 can also participate in the regulation of histone modifications[37,38]. ChIP-PCR showed that the dimethylation (H3K9me2) and trimethylation (H3K9me3) of H3K9

were increased in *Klf4*, *PU.1*, and *Notch1* genes in HSCs from T2D mice. When Dnmt1 was inhibited, these repressive modifications at H3K9 were reduced, and the expression of these genes was restored in HSCs from T2D mice. These findings indicate that the upregulation of Dnmt1 increased the repressive histone modifications in *Notch1*, *PU.1*, and *Klf4*, thereby inhibiting the expression of these genes in HSCs from T2D mice.

Moreover, the identification of the let-7d-3p-dependent regulation of Dnmt1 in type 2 diabetic HSCs, which regulates the genes responsible for HSC differentiation towards monocytes as well as their M1/M2 polarization, creates an excellent opportunity for the development of biological therapeutics to restore normal wound healing in people with T2DM. In addition, two independent studies show that the let7 family modulates angiogenesis and contributes to the dysfunction of *db/db* type 2 diabetic bone marrow-derived angiogenic cells[43,44]. We now show the mechanism responsible for these findings.

Finally, based on in vitro experiments, we showed that normal human HSCs have the same response to hyperinsulinemia-induced Nox-2-dependent oxidant stress. Indeed, hyperinsulinemia increased the expression of Nox-2 and Dnmt1 in human HSCs. This led to a reduction of human HSC differentiation towards macrophages and skewed human macrophages towards M1 polarization.

Taken together, these insights into the mechanism by which T2DM impairs wound healing open multiple avenues to new

promising techniques to restore normal wound healing and thereby reduce the risk of amputation in people with T2DM.

## Methods

**Animals**. In total, 8–12 weeks old male C57BL/6J wild type (WT) and B6.BKS(D)-Lepr[db]/J (db/db) mice were used in the studies. All mice were purchased from Jackson Laboratories and were maintained in mouse barrier facilities. Care and use of mice was in accordance with NIH guidelines and the Institutional Animal Care and Use Committee of the University of Massachusetts Medical School approved all protocols and granted us permission to perform all the experiments described in this manuscript. db/db and WT mice were fed standard chow diet (5.4 g/100 g diet, 0% cholesterol) and HFD mice were generated by feeding C57BL/6J mice a HFD (60% fat calories; Research Diet, New Brunswick, NJ). NAC was given for 8 weeks (150 mg/kg/day via drinking water).

**Induction of cutaneous wounds**. Cutaneous wounds were induced as described in Luo et al.[14]. In brief, mice were anesthetized using isoflurane. The dorsal surface was shaved, washed with povidone-iodine solution and cleaned with an alcohol swab. Two wounds, one on each side of the midline, were patterned on the shaved dorsum of the mice using a sterile 6-mm punch biopsy tool (Miltex, Inc. York, PA 17402, USA). A silicone stent was secured around the perimeter of the wound. To improve adherence of the wound dressing, tincture benzoin was applied to the perimeter of the wound and allowed to dry. Finally, the wound was covered with a transparent, bio-occlusive dressing (Opsite, Smith and Nephew Medical Limited, Hull, England) thereby creating a moist wound chamber environment. Following surgery, the mice were placed in individual cages, and allowed to fully recover from the anesthesia. The animals were housed in the institutional animal facility.

**Morphometric analysis of wounds**. Wound closure rate was calculated by macroscopic quantification and histological analysis. For macroscopic quantification, wound photographs were taken with a Nikon camera at the indicated time points and the wound areas were measured by imageJ software. For histological analysis, 5 μm paraffin sections were stained with hematoxylin–eosin (H&E). Images were taken using a NikonEclipse 90i microscope. The distances between the epithelial tips/the edges of panniculus carnosus and granulation tissue were measured by NIS-Element AR analysis.

**Immunohistochemistry of wounds**. Immunohistochemical staining was performed on 5 μm cryosections of day 7 wounds. Briefly, 5 μm cryosections were fixed either in 4% paraformaldehyde (CD144 and SMC vessels) or in acetone (F4/80 macrophages) and blocked with 5% goat serum. For macrophage staining primary antibodies, we used rat mAbs against F4/80(1:40, AbD serotec MCA497G, clone:CI: A3-1), rabbit mAbs against iNOS (1:800, Novus biologicals NBP1-33780, clone: K13-A), rat mAbs against TNFα (1: 10, AbD serotec MCA1488, clone: MP6-XT22), rabbit pAbs against CD206 (1:400, Santa Cruz sc-48758), and rabbit pAbs against Arginase (1:400, Novus biologicals NBP1-32731). For secondary antibodies, we used Alexa Fluor 488 goat anti-rat IgG(1:500, Invitrogen A11006), Alexa Fluor 555 goat anti-rat IgG(1:500, Invitrogen A21434), or Alexa Fluor 555 goat anti-rabbit IgG (1:500, Invitrogen A21428). For vessel staining, we used CD144 (1:40, BD pharmingen 550548) and FITC conjugated α-smooth muscle actin (α-SMA, 1:2000, Sigma-Aldrich F3777, clone:1A4). Immunofluorescent images were taken using a Zeiss microscope axiocam at ×200 magnification.

**Hematopoietic stem cell sorting and transplantation**. Whole bone marrow cells were flushed out from femurs and tibialis. Bone marrow cells were stained with antibodies for the identification of HSCs (c-Kit+Sca1+lineage-CD90.1[lo/−]). The antibodies used were: c-Kit (1:50, eBioscience 12–1171, clone: 2B8), Sca-1 (1:50, eBioscience 11–5981, clone: D7), CD90.1 (1:50, eBioscience 45–0900, clone:HIS51) and a lineage cocktail antibody (1:50, BD Bioscience 558074). As for the gating strategy for flow cytometry, we first gated on single, viable cells and eliminated any debris, dead cells and clumps or doublets. We then drew gates on the c-Kit-positive/Sca1-positive/lineage-negative/CD90.1[low]-negative population for HSC sorting. Cell sorting was performed on a MoFlow or FACSAria cell sorter.

To reconstitute the hematopoiesis, the recipient mice were lethally irradiated (with a split dose of 1100 Gy) and 3,000 HSCs were transplanted to each recipient by retro-orbital injection. For lentiviral-shRNA transfected HSCs, around 10,000 cells were transplanted to each recipient.

**Lentivirus conjugated shRNA transduction**. Lentiviral-shRNA constructs were tagged with GFP and purchased from the RNAi core facility at UMASS Medical School. Lentiviral particles were generated in 293 T cells, which were purchased from ATCC (CRL-3216). Sorted HSCs were incubated with lentiviral supernatant with 8 μg/ml polybrene at 37 °C for 20–30 min and then spun down at 25 °C, 3000 rpm for 1.5 h. The supernatant was then replaced with lentiviral supernatant and HSC medium with 4 μg/ml polybrene in 6-well plates and incubated overnight. The medium was replaced after 24 h and the cells were cultured for an additional 1–2 days. Finally, GFP positive cells were purified by FACS sorting.

**HSC-induced differentiation towards macrophages**. HSCs were cultured in complete RPMI 1640 medium (20% FBS, 100 U/ml penicillin, 100 U/ml streptomycin, 50 μg/ml beta-mercaptoethanol, 1% Glutamax (GIBCO), 1% nonessential amino acid (GIBCO), and 1% sodium pyruvate(GIBCO)) supplemented with 50 ng/ml SCF, 10 ng/ml TPO, and 10 ng/ml Flt3 with either 6 ng/ml IL-3 and 10 ng/ml IL-6, or 10 ng/ml M-CSF or 40 ng/ml M-CSF (Peprotech). After 6 days, cells were induced to differentiate towards M1/M2 macrophage by changing the medium to the M1 induction medium (HSC basic medium, 5% FBS, 50 ng/ml LPS and 5 ng/ml IFNγ) or the M2 induction medium (HSC basic medium, 5% FBS, 10 ng/ml IL-4) overnight. Cells were then collected and stained with macrophage markers (F4/80 (1:50, eBioscience 12–4801, clone:BM8), CD115(1:50, eBioscience 17-1152-82, clone: AFS98), CD11b (1:50, eBioscience 45-0112-82, clone:M1/70)), M1 macrophage markers (TNFα(1:50, eBioscience 17-7321-81,clone:MP6-XT22), iNOS (1:50, BD Bioscience 610331), IL-12(1:50, eBioscience 45-7123-80, clone: C17.8)), or M2 macrophage markers (Arginase 1(1:400, Novus biologicals NBP1-32731), CD206(1:50, Biolegend 141708, clone:C068C2), IL-10(1:50, eBioscience 11-7101-82, clone:JESS-16E3)). Stained cells were run through a BD FACSCalibur flow cytometer and data were analyzed using the FlowJo software. All the cytokines were purchased from Peprotech and all the antibodies were purchased from BD Biosciences.

**Macrophage concentration and M1/M2 macrophage identification**. Wound cutaneous samples were collected by 2 mm biopsy punch and minced to 2 mm$^2$ sections on ice by digestion with dispase (Roche, overnight at 4 °C), and hyaluronidase (Sigma) and Collagenase (Roche, 2 h at 37 °C). Single-cell suspensions (10$^6$ cells/μl) were incubated with rat anti-mouse CD16/CD32 antibody(1:50, eBioscience 14-0161-82, clone:93) at 4 °C for 15–30 min to minimize nonspecific staining of leukocyte Fc receptors. After incubation, cells were washed once and resuspended in FACS buffer for antibody staining. For macrophage identification, we used F4/80, CD115, and CD11b antibodies. For M1 macrophage identification, we used TNFα, iNOS, and IL12 antibodies. For M2 macrophage identification, we used Arginase 1, CD206, and IL10 antibodies. The source and dilution of all the antibodies were the same as described above. Stained cells were processed on a BD FACSCalibur flow cytometer and analyzed using FlowJo software.

**Quantitative real-time PCR**. RNA was extracted from cells using RNAqueous Micro Kit as directed by the manufacturer (Ambion). cDNA synthesis was performed using Superscript III reverse transcriptase as directed by the manufacturer (Invitrogen). Quantitative real-time PCR (qRT-PCR) was performed using SYBR Green Mix (Bio-Rad) on an Eppendorf Master Cycler. Primers are listed in Supplementary Tables 1 and 2.

**Western blot analysis**. Cells were homogenized in lysis buffer (50 mM N-2-hydroxyethylpiperazine-N′-2-ethanesulfonic acid [pH 7.5], 150 mM magnesium chloride, 1 mM ethylenediaminetetraacetic acid, 100 mM sodium chloride, 1% NP40). Protein extracts underwent sodium dodecyl sulfate polyacrylamide gel electrophoresis (SDS-PAGE) and were transferred to a nitrocellulose membrane. The membrane was blocked with 5% milk for 1 h at RT followed by incubation with one of the following primary antibodies overnight at 4 °C: Dnmt1 (1:1000, Sigma D4567), Dnmt3a (1:1000, ThermoFisher MA1-91490, clone:64B814), Dnmt3b (1:1000, ThermoFisher Scientific PA1-884), and α-tubulin(1:5000, Sigma T6074, clone:B-5-1-2). Following washes, membranes were incubated with a specific horseradish peroxidase-conjugated secondary antibody which include anti-mouse IgG (1:3000, Promega W4021) and anti-rabbit IgG (1:3000, Promega W4011) for 1 h at room temperature and proteins were visualized by incubation with ECL (ThermoFisher Scientific). Quantification was performed using ImageJ software. Uncropped versions of all Western Blots can be found in Supplementary Figure 17.

**Pyrosequencing**. Genomic DNA was isolated by traditional phenol–chloroform extraction and isopropanol precipitation. DNA concentration was measured using a NanoDrop spectrophotometer. Around 500 ng genomic DNA were bisulfite converted by EZ DNA methylation-Gold kit following the manufacturer's instructions (Zymo Research). PCR amplification was performed using 2xHiFi Hotstart Uracil + ready mix PCR (Kapa Biosystems). PCR and pyrosequencing primer sets with one biotin-labeled primer were used to amplify the bisulfite converted DNA. Primers were designed using PyroMark Assay Design software version 2.0.1.15 (Qiagen). The size of the amplicons was 200 bp or less. Briefly, 5 μl master mix, 5 pmol of each primer, 20 ng genomic DNA and ultra-pure water to a final volume of 10 μl were mixed for each reaction and run at thermal cycling conditions: 95 °C for 3 min and then 50 cycles: 20 s at 98 °C; 15 s at the optimized primer-specific annealing temperature; 15 s at 72 °C and a final extension for 1 min at 72 °C. The amplified DNA was confirmed by electrophoresis on a 2% agarose gel. 2 μl streptavidin beads (GE Healthcare, Buckinghamshire, UK), 40 μl PyroMark-binding buffer, 10 μl PCR product and 28 μl water were mixed and incubated for 10 min on a shaking platform at 1300 rpm. Using the Biotage Q96 Vaccum Workstation, amplicons were separated, denatured, washed, and added to 25 μl annealing buffer containing 0.33 μM of pyrosequencing primer. Primer annealing was performed by incubating the samples at 80 °C for 2 min and allowed to cool to

room temperature prior to pyrosequencing. PyroGold reagents were used for the pyrosequencing reaction and the signal was analyzed using the PSQ 96MA system (Biotage, Uppsala, Sweden). Target CGs were evaluated by instrument software (PSQ96MA 2.1), which converts the pyrograms to numerical values for peak heights and calculates the methylation at each base as a C/T ratio. Primers are listed in Supplementary Table 3.

**ChIP-PCR.** $1 \times 10^6$ HSCs were sorted and crosslinked with 1% formaldehyde at room temperature for 10 min, and the reaction was stopped by adding Glycine to a final concentration of 0.125 M and incubating at room temperature for 5 min. Crosslinked cells were lysed in lysis buffer and sonicated to 200–500 bp fragments (Bioruptor, Diagenode). The sonicated chromatin was centrifuged at 4 °C for 5 min. The crosslinked DNA was immunoprecipitated with histone H3 pan, H3K9me3, H3K9me2, or H3K9Ac antibodies (Millipore, USA) overnight at 4 °C with rotation, DNA-Antibody complexes were bound to ChIP beads, pulled down, washed, and then eluted from the beads. Following reversal of cross-linkage, purified DNA was used for quantitative PCR using ChIP-PCR primers purchased from IDT (MA, USA). Immunoprecipitation efficiency was calculated by normalizing sample $C_T$ values against control IgG values and calculating ratios of sample $C_T$ values relative to histone H3 values. Primers are listed in Supplementary Table 4.

**MiRNA microarray expression profiling.** HSCs were isolated as described. Total RNA was isolated using the mirVana miRNA isolation kit according to the manufacturer's instructions (Applied Biosystems). MicroRNA expression was measured by Affymetrix miRNA 3.0 array (Affymetrix, Santa Clara, CA, USA). The sample labeling, microarray hybridization, and washing were performed based on manufacturer's standard protocols. The slides were washed and stained and the arrays were scanned by the Affymetrix Scanner 3000 (Affymetrix). The scanned images were analyzed using Expression Console software (version 1.3.1 Affymetrix). The CEL files generated from the assays were analyzed using the Affymetrix Expression Console Software. Based on our microarray results and a review of the literature, candidate miRNAs were chosen for further validation with qRT-PCR. All experiments were conducted in the Genomic Core Facility of the University of Massachusetts Medical School.

**qRT-PCR analysis of microRNA expression.** Total RNA was isolated using the mirVana miRNA isolation kit (Applied Biosystems). cDNA was synthesized using the QuantiMir RT Kit small RNA quantification system (SBI). Mouse U6 was used as an endogenous control. The let-7d-3p mimic and inhibitor were purchased from Life Technologies (catalog numbers 4464066 and 4464084, respectively).

**miRNA target prediction.** The predicted target genes of differentially expressed miRNAs were obtained using the following tools: TargetScan v6.2, miRDB, and Affymetrix Expression Console Software. The search was performed on the 3′-UTR regions of target mRNAs with a $p$ value of 0.05 defining the probability distribution of random matches set in the software with a minimum miRNA seed length of 7.

**mRNA 3′-UTR cloning and luciferase reporter assay.** HEK293T cells grown in 96-well plates were transfected with pmirGLO containing the Dnmt1 3′ UTR region that includes the let-7d-3p-binding sites. The cells were co-transfected with let-7d-3p using Lipofectamine (Invitrogen). The Firefly and Renilla luciferase activities in the cell lysates were assayed with a Dual-Luciferase Reporter Assay System (Promega) at 48 h post-transfection. To generate the mutant variants, point mutations in the binding sites of let-7d-3p in the 3′-UTR region of Dnmt1 were introduced by PCR according to the site-directed mutagenesis protocol from Agilent. The two mutants that were generated are labeled "Dnmt1 M1" and "Dnmt1 M2".

**Analysis of intracellular ROS by DCF-DA staining.** Cells were resuspended in PBS and incubated with 5 µM DCF-DA (Sigma) for 30 min at 37 degrees in the dark. Cells were then washed twice with PBS and analyzed on a BD FACSCalibur analyzer (excitation wavelength 492–495 nm; emission wavelength 517–527 nm). The resulting data was analyzed using FlowJo software.

**Dot blotting.** Dot blotting was performed following the method described in Hainer et al.[45]. Briefly, serial dilutions of genomic DNA starting at 300 ng were denatured at 95 °C for 10 min, then put on ice immediately for 10 min. Denatured DNA samples were spotted onto Amersham Hybond N + nylon membrane (GE Healthcare, Uppsala, Sweden) and the membranes were UV crosslinked. The membrane was incubated in 0.1% SDS overnight, blocked with 5% nonfat milk and 3% BSA for 2 h, incubated with anti-5mC (1:1000, Eurogentec BI-MECY, RRID: AB_2616058) for 1 h, washed three times with PBS-T, incubated with HRP conjugated anti-mouse secondary (1:10,000, Bio-Rad, Hercules, CA, USA, 170–6516, RRID:AB_11125547) for 1 h, washed three times with PBS-T, and detected with enhanced chemiluminescence. For loading, gDNA samples were diluted simultaneously, spotted directly onto Amersham Hybond N + nylon membrane (GE

Healthcare) and the membranes were UV crosslinked. Membranes were incubated with 0.2% methylene blue for 5 min and washed five times with water.

**Human HSCs.** Fresh whole human bone marrow was purchased from ALL Cells (1301 Harbor Bay Pkwy #200, Alameda, CA 94502). Mononuclear cells were isolated with Ficoll-Paque PLUS (1.077 density) and CD34 + cells were sorted and cultured in RPMI 1640 supplemented with cytokines (50 ng/ml SCF, 50 ng/ml Flt3, 20 ng/ml TPO, 20 ng/ml IL-3, 20 ng/ml IL-6) with or without 174 nM (equal to 1.0 µg/ml) insulin for 48 h. For induced differentiation towards macrophages, $1.0 \times 10^5$ HSCs were seeded in 6-well plates with 1.5 ml medium containing 25 ng/ml M-CSF for 3 days. Fresh medium was added every other day. For M1/M2 differentiation, on day 3, cells were washed with PBS and RPMI1640 containing 5% FBS, 1% PS, 100 ng/ml IFNγ (for M1), or 10 ng/ml IL-4 and IL-13(for M2) was added for 24 h. The human macrophage markers used are: CD68(1:50, eBioscience 12-0689-41, clone:eBioY1/82 A), CD14(1:50, eBioscience 11-0149-42, clone:61D3), CD206(1:50, eBioscience 12-2069-41, clone:19.2), CD163 (1:50, eBioscience 17-1639-41, clone: eBioGHI/61).

**Statistical analysis.** All data are shown as means ± SEM. Statistical analyses were carried out with GraphPad Prism (GraphPad Software) software. Statistical significance was evaluated by using a one way analysis of variance (ANOVA) or an unpaired $t$ test. Significance was established for $p$ values of at least <0.05.

**Data availability.** The data discussed in this study have been deposited in NCBI's Gene Expression Omnibus which is accessible through the GEO series accession number GSE89842. All data that support the findings of this study are available from the corresponding author upon request.

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

## Acknowledgments

This work was supported by the Wayne and Gladys Valley Foundation (to L.M.M.). We thank Dr. Oliver Rando (University of Massachusetts Medical School) for his great support with epigenetic studies. We thank Phyllis Spatrick (Genomic Core Facility of University of Massachusetts Medical School) for the conduction of microRNA micro-array analysis.

## Author contributions

J.Y., G.T., and L.M.M. designed all experiments. J.Y. and G.T. performed and analyzed all experiments. S.W. assisted with animal care, wounding model and qRT-PCR. A.T. assisted with animal care and pyrosequencing. L.K. assisted with microRNA experiments. N.D. provided assistance with experimental design and revision of figures. T.G.F. provided assistance with experimental design and revision of the manuscript. J.Y., G.T., L.K., and L.M.M. wrote the manuscript.

## Additional information

**Competing interests:** The authors declare no competing financial interests.

