## [Peer Review File · Nature Communications]

Reviewers' comments:

Reviewer #1 (Remarks to the Author):

“ Diabetes Impairs Wound-healing by DNMT1-dependent Dysregulation of Hematopoietic Stem Cells Differentiation towards Macrophages”

Overview

The manuscript by Yan et al. analyzes the important problem of impaired wound healing in diabetes. Their finding about the change in M1 and M2 macrophages, effects of hyperinsulemia, and oxidative stress are very interesting, however, the claim that impaired wound healing is due to DNMT1-dependent dysregulation is not justified by the data in the manuscript. Authors should re-do methylation analysis, verify ChIP analysis and show more data for DNMT1 protein levels to make a convincing conclusion. The major points below should be absolutely addressed, particularly methylation analysis of Notch 1, PU.1 and KLF4.

Major points

- 1) Authors show that Dnmt1 expression is increased approximately 2-fold in db/db versus WT mice (Sup. Fig. 8b). Does this translate into the increase of DNMT1 protein? Authors should show a Western blot. In addition, does this translate into any global changes of DNA methylation?
- 2) Authors show that shRNA knockdown of Dnmt1 results in wild-type expression levels (Sup. Fig. 8b). Does this result in lower production of DNMT1 protein? Authors should show a Western blot.
- 3) Methylation analysis of Notch1, PU.1 and KLF4 has many problems. Authors have to show the CpG site density of analyzed sequences (PCR amplified fragments) of Notch 1, PU.1 and KLF4. Changes in DNA methylation are only relevant for CpG-rich promoters. Currently, we don't know how many CpG sites are analyzed for each gene. If the total number is five and only one site loses methylation that would result in 25% decrease in methylation as detected by pyrosequencing but it's biologically meaningless. Authors should re-do methylation analysis using bisulfite sequencing and results should be displayed with methylation data for individual sites. This is essential for the claim of the manuscript that impaired wound healing is due to the increased methylation of Notch 1, PU.1 and KLF4.
- 4) Fig. 4 c-f plots seem to be mislabeled, as the labels don't correspond to the figure legend. Consequently, it's hard to interpret these data with confused labels.
- 5) How can authors explain that expression of PU.1 is not affected in db/db+shDNMT1 (Sup. Fig. 8b) while they show that PU.1 methylation levels are changing in Fig. 6a, and, in fact, exhibit the greatest drop in methylation as compared to Notch 1 and KLF4? (PU.1 ~ 28% less methylation in db/db+shDNMT1 vs db/db; Notch 1 ~ 24%; KLF4 ~ 19%).
- 6) ChIP-PCR analysis of histone modifications has to be improved. First, fold enrichment of 1-1.4 in relation to IgG is a very low enrichment that seems like background. Second, ChIP should be performed against unmodified histone H3 to normalize the number of nucleosomes. Third, authors should test known regions that contains high levels of H3K9me2/3 modifications or are known to be devoid of these modifications in order to control for there IP and qPCR.

Minor points

- 1) Explain in the text Fig. 4b: what is DNMT1, DNMT1 M2 and DNMT1 M3.
- 2) On p. 5 it says “DNA methyltransferase 1 (DNMT1), a key enzyme mediating DNA methylation and histone modifications.” DNMT1 does not modify histones. There are many other misleading statements throughout.

Reviewer #2 (Remarks to the Author):

In the current study by Yan et al. they found out that impaired wound healing in diabetic mice is mediated by an HSC-autonomous mechanism. HSCs under diabetic conditions show an increased oxidative stress response that is leading to the de-repression of the let-7 target gene DNMT1. Upregulation of DNMT1 in turn downregulates some of its target genes important for HSC differentiation towards macrophages. Finally, this dysregulation within HSCs in diabetic mice reduces macrophage infiltration and stimulates M1 polarization within the wound area. The study is interesting and the understanding of the regulation of the inflammatory response in wound healing is of high relevance. Nevertheless, there are several major technical concerns prior publication, because the manuscript is too preliminary at its present state.

Specific points:

1. It is well established that the wound healing response in db/db mice is delayed in comparison to wild type mice (Figure 1). But the finding that HSCs isolated from db/db mice delay wound healing in irradiated wild type mice and that wild type HSC rescue the wound healing response in db/db mice is essential for the current study, but only superficially analyzed. The authors should provide a detailed analysis of the wound healing response with histological data. The macroscopic quantification of wound closure is not sufficient to assure improved or delayed healing. This is particularly relevant since no differences can be seen between the representative images of wt->wt and db/db->wt transplanted mice (in Figure 5b) and between wt->db/db and db/db-> db/db mice (in suppl Figure 6b). For the three studies described in Figure 2, 5 and Supp Figure 6 the authors should provide 1) quantification for the wound healing parameters distance between epithelial tips and amount of granulation tissue based on histological images of wound tissue isolated at day 7, 14 and 21 post injury, 2) provide histological findings to illustrate the differences in macrophage infiltration and macrophage polarization and 3) determine vascularization of the wounds. Furthermore, the authors should add a scale bar in their images shown in Figure 1b and 5b, because it appears that the distance varies. In Figure 2 the representative macroscopic images are additionally missing and should be added.
2. The authors should add their raw data (dot plots) for the FACS analysis in Figure 1e-g, 2d-f and 5d-f including the percentage of all cell populations. Since it is not mentioned if the authors used any isotype-matched controls for their FACS analysis, they should provide some data proving the specificity of their antibodies.
3. Regarding Figure 3, at no point in the manuscript it is explained what DCF+ cells are and how the experiment was done. The NAC treatment is missing in the material and method section, too. The authors should provide the missing information.
4. It is unclear how microRNAs shown in suppl figure 4 have been selected. Since many let-7 family members share the seed sequences, the regulation of family members should be shown in detail. Moreover, the authors should comment on the targets of let-7-5p which appears up-regulated.
5. The differences in DNA methylation is modest (max 10 %). Is this sufficient to change gene expression? The data should be provided in more detail (% methylation of the specific cytosines at the promoters, TF binding sites should be shown).
6. What is the effect of let-7 on DNA methylation and histone modifications?
7. The authors should provide the corresponding Western Blot analysis for the immunoprecipitation of H3K9me3, H3K9me2 and H3K9Ac.
8. The manuscript is not very well written: the introduction should end with the aim of the study, instead of recapitulating the authors own previous studies (should be included in state of the art). Also in the result section it should be better explain why the next steps were done (e.g. the regulation of redox related enzymes and next the elucidation of Dnmt1 comes without a logical link.
9. The literature is also not adequately covered. Previous studies report already that let-7 affects bone marrow cell functions in dbdb mice (Bae et al Arterioscler Thromb Vasc Biol. 2013

Aug;33(8):1920-7) and let-7 has additional functions in angiogenesis (e.g. Kuehbacher et al Circ Res. 2007 Jul 6;101(1):59-68).

Reviewer #3 (Remarks to the Author):

This is a very interesting study demonstrating that wild type mice transplanted with HSC from T2D mice have impaired cutaneous wound healing, similar to what is observed in T2D mice. Impaired wound healing was associated with reduced monocyte/macrophage abundance in wounds, and increased M1 macrophage polarisation, as is observed in poorly healing wounds in patients with T2D. HSC from T2D mice were impaired in their differentiation along the monocyte/macrophage lineage. The mechanism underlying reduced differentiation and increased M1 polarisation was related to increased oxidative stress in T2D HSC (possibly due to hyperinsulinemia), which was found to reduce expression of Mir- let-7d-3p, which in turn increased expression of the let-7d-3p target, DNMT1. Increased DNMT1 was shown to have a causative role in the impaired wound healing, as DNMT1-deficient T2D HSC transplanted into wt mice rescued the impaired wound healing associated with T2D HSC transplant. Further, DNMT1-deficient T2D HSC could improve wound healing in T2D mice. Mechanistically, DNMT1 was implicated in repressive chromatin modification (methylation, histone acetylation) at several loci involved in macrophage differentiation and polarisation, specifically PU.1, Notch1 and KLF4. This is a novel mechanism by which T2D leads to epigenetic reprogramming of HSCs, that appears to have flow on consequences for monocyte/macrophage differentiation and wound healing – although the concept itself is not entirely novel (authors' reference 8 demonstrated a similar phenomenon with respect to wound healing in T2D).

Specific comments/questions

- Re monocyte/macrophage phenotyping, does the f480+/cd115+/cd11b+ phenotype include all monocytes/macrophages in bone marrow and in the wound? – Resident macrophages, in both bone marrow and peripheral tissues, are frequently cd115low/neg? This information is critical to the claims of 'reduced monocyte/macrophage differentiation' and 'infiltration' that are made throughout the manuscript. Flow gating strategies for marrow and wound populations should be shown, including M1/M2 macrophage phenotyping.
- Is there a potential role for host tissue resident macrophages in healing wounds in HSC-transplanted mice? Ie cell intrinsic effects from the db/db HSC may influence local macrophage populations during healing?
- Flow cytometry data is expressed as cell proportions – absolute cell numbers would be valuable to confirm whether changes in cell proportion relates to altered cellularity in the marrow of wt v db/db mice. E.g. in the Discussion it is claimed that "the total number of macrophages in wounds of T2D mice is significantly lower than that in WT mice", which is not shown in the data presented. Is the reduction in monocytes/macrophages in db/db bone marrow associated with expansion of precursor cells, or another lineage that may be detrimental to wound healing – eg granulocytes?
- The hypothesis is that sustained elevated DNMT1 expression underpins the impaired differentiation of HSC along the macrophage lineage, as well as macrophage migration and maturation/polarisation in the tissue microenvironment. Is there elevated DNMT1 expression in db/db bone marrow or wound monocytes/macrophages; or is it just in HSC? i.e. although there are fewer of them, are the cells that are able to differentiate, able to overcome the repressive effect of DNMT1?
- The data transformation and presentation as "% change" used throughout for the M1/M2 macrophage profiling (eg Fig 1e-f) seems to me to lose valuable information. In the current presentation, it is not possible to discern the % of monocytes/macrophages that express an M1/M2 phenotype, how this changes during wound healing in the wt setting, and how it is affected in the diabetic or manipulated setting. In addition, there no error bars on these figures, and it is not clear whether the data represent fluorescence intensity, or % positive cells, or something else? Was any co-staining for these markers performed?

- In Figure 4b – what are M1 and M2 – presumably mutated MiRNA binding sites, but this is not clear in methods or figure legend. Contrary to the results text only one of them affects reporter expression.
- Figure 4c-f: Are some of these panels mislabelled, they do not correspond to the results text/figure legend? According to the legend, 4c-d represent DNMT1 expression in wt cells, not let7 expression as on y axis in panel c. Why would a let7 mimic decrease native miRNA expression in wt cells as shown in c? Similarly with e-f – the let7 mimic increased let7 expression, and the inhibitor decreased DNMT1 expression in db/db cells? There are no details of these MiRNA mimics or inhibitors in the methods section.
- The selected macrophage differentiation factors PU.1, Notch1 and KLF4 play important roles in different phases of monocyte/macrophage differentiation – I was curious about the rationale for the selection of these particular genes, which all turned out to be epigenetically modified? Were other candidate genes investigated? – how specific is DNMT1 likely to be, how widespread the hypermethylation effect etc could be discussed.
- I felt the discussion offered little more than a restatement of results. The comment in the introduction “This potentially novel mechanism may be responsible for the conflicting evidence existing between the observations made at different stages of wound healing in T2DM patients and in animal models.” should be explained/further discussed as relevant to the results of this study. Potential triggers for oxidative stress, especially NOX2, could be discussed – is there a link between hyperinsulinemia and NOX2?
- Methods:
- Murine HSC cultures were “supplemented with 50ng/ml SCF, 10ng/ml TPO and 10 ng/ml Flt3 with either 6ng/ml IL-3 and 10ng/ml IL-6 or 10ng/ml M-CSF or 40ng/ml M-CSF” In what circumstances were il6/il3 used vs M-CSF and what is the relevance of this? It is then stated that “After 6 days, cells were induced to differentiate towards M1/M2 macrophage by changing the medium to the M1 induction medium (HSC basic medium, 5% FBS, 50ng/ml LPS and 5ng/ml IFN γ) or the M2 induction medium (HSC basic medium, 5% FBS, 10ng/ml IL-4) overnight.” – were in vitro polarisation experiments on BM HSC performed, as this suggests?
- Human HSC culture – how long was the differentiation in insulin prior to mono/mac differentiation? KLF4 expression was downregulated in response to insulin in human HSC – what happened to the expression of PU.1 and Notch1?

Point-by-point response to reviewers

We thank all the reviewers for their constructive and insightful comments, which have certainly helped us improve the quality of our manuscript. In addition, we sincerely appreciate the reviewers highlighting the major novel findings of our study.

Reviewer #1

Overview

The manuscript by Yan et al. analyzes the important problem of impaired wound healing in diabetes. Their finding about the change in M1 and M2 macrophages, effects of hyperinsulinemia, and oxidative stress are very interesting, however, the claim that impaired wound healing is due to DNMT1-dependent dysregulation is not justified by the data in the manuscript. Authors should re-do methylation analysis, verify ChIP analysis and show more data for DNMT1 protein levels to make a convincing conclusion. The major points below should be absolutely addressed, particularly methylation analysis of Notch 1, PU.1 and KLF4.

Major points

- 1) Authors show that *Dnmt1* expression is increased approximately 2-fold in *db/db* versus WT mice (Sup. Fig. 8b). Does this translate into the increase of DNMT1 protein? Authors should show a Western blot. In addition, does this translate into any global changes of DNA methylation?

As requested by the reviewer, we have performed Western blots for multiple experiments on protein extracts from WT and *db/db* HSCs to measure the expression of DNMT1 at the protein level. These blots are shown in revised **Supplementary Figure 9a,b and Figure 4h** of the manuscript. These results show that oxidant stress dependent upregulation of the *Dnmt1* gene results in a concomitant increase in DNMT1 protein in *db/db* HSCs.

For global changes in DNA methylation, we did a dot blot analysis of 5mC that demonstrates a global increase in DNA methylation in *db/db* HSCs relative to WT HSCs. These results are shown in revised **Supplementary Figure 7e** of the manuscript.

- 2) Authors show that shRNA knockdown of *Dnmt1* results in wild-type expression levels (Sup. Fig. 8b). Does this result in lower production of

DNMT1 protein? Authors should show a Western blot.

As requested by the reviewer, we have performed a Western blot on protein extracts from WT, *db/db* and *db/db* +shDNMT1 HSCs to look at DNMT1 protein levels. The blot is shown in revised **Supplementary Figure 9a,b** of the manuscript. ShRNA knockdown of DNMT1 in *db/db* HSCs significantly decreased DNMT1 protein levels. We have modified the text of the manuscript on page 12 to reflect these results.

3) Methylation analysis of Notch1, PU.1 and KLF4 has many problems. Authors have to show the CpG site density of analyzed sequences (PCR amplified fragments) of Notch 1, PU.1 and KLF4. Changes in DNA methylation are only relevant for CpG-rich promoters. Currently, we don't know how many CpG sites are analyzed for each gene. If the total number is five and only one site loses methylation that would result in 25% decrease in methylation as detected by pyrosequencing but it's biologically meaningless. Authors should re-do methylation analysis using bisulfite sequencing and results should be displayed with methylation data for individual sites. This is essential for the claim of the manuscript that impaired wound healing is due to the increased methylation of Notch 1, PU.1 and KLF4.

Thank you for the reviewer's insightful comments. To address this concern, we have repeated our pyrosequencing experiments with several more CpGs and have now plotted our data to show methylation changes at individual CpG sites, which is shown in revised **Figure 8a and Supplementary Figure 13** of the manuscript. In revised **Figure 8a** of the manuscript, we show significant methylation changes for 6 out of 8 CpGs for Notch1, 5 out of 11 CpGs for PU.1 and 11 out of 12 CpGs for KLF4.

We believe that, as presented, the results now show convincingly the significant methylation changes and reinforce our claim that impaired wound healing is due to the increased methylation of Notch 1, PU.1 and KLF4.

A major advantage of pyrosequencing is that it allows the direct sequencing and methylation analysis of a PCR product in a quantitative manner without requiring dozens of bacterial clones to be sequenced for each genotype in each experiment, which would be required when the differences in methylation are in the range of 10-50% at most CpGs tested.

Pyrosequencing is a state-of-the-art method to reliably detect hypomethylation, hypermethylation and mixed methylation of DNA in a

cost-effective, quantitative manner with reduced bias and a lower workload. It is a well-validated method with a real-time quantitative read-out that is highly suitable for sequencing short stretches of DNA.

In the previous version of the manuscript, we averaged CpGs from each gene together to save space, but as the reviewer rightly noted, the previous figure left out important information about the numbers of CpGs tested and the effect on each. However, the pyrosequencing method does measure methylation of individual CpGs, just as sequencing of individual clones does, and provides this information in a quantitative manner. The new **Figure 8a and Supplementary Figure 13** address all of these concerns and show significant increases in DNA methylation at all genes tested in *db/db* HSCs.

4) Fig. 4 c-f plots seem to be mislabeled, as the labels don't correspond to the figure legend. Consequently, it's hard to interpret these data with confused labels.

We apologize for the confusion and have relabeled the plots and adjusted the figure legend in order to clearly explain plots c-f in revised **Figure 5c-f** of the manuscript.

5) How can authors explain that expression of PU.1 is not affected in *db/db+shDNMT1* (Sup. Fig. 8b) while they show that PU.1 methylation levels are changing in Fig. 6a, and, in fact, exhibit the greatest drop in methylation as compared to Notch 1 and KLF4? (PU.1 ~ 28% less methylation in *db/db+shDNMT1* vs *db/db*; Notch 1 ~ 24%; KLF4 ~ 19%).

Thank you for the reviewer's insightful comments. In the original version of the manuscript, the expression of PU.1 in *db/db+shDNMT1* HSCs showed a slight increase, but the change did not reach statistical significance (n=6). To address this concern, we repeated our qPCR analysis using the same primer pair 5 more times for a total number of n=11. In doing so, our results reached statistical difference between *db/db* HSCs and *db/db+shDNMT1* HSCs as shown below (p=0.039).

In addition, in order to confirm this result, we measured the expression of PU.1 in all three groups with a second set of primers (P2). The locations of the primers are listed below. Gene expression analysis with the new primers showed an almost 2-fold increase in PU.1 in the *db/db+shDNMT1*, compared to the *db/db* sample, with a p-value of 0.01 (revised **Figure 7b** of

the manuscript). Taken together, we believe that these results confirm that the decrease in PU.1 expression in *db/db* HSCs is rescued when DNMT1 is silenced, which correlates well with the methylation changes.

6) ChIP-PCR analysis of histone modifications has to be improved. First, fold enrichment of 1-1.4 in relation to IgG is a very low enrichment that seems like background. Second, ChIP should be performed against unmodified histone H3 to normalize the number of nucleosomes. Third, authors should test known regions that contains high levels of H3K9me2/3 modifications or are known to be devoid of these modifications in order to control for there IP and qPCR.

Thank you for the reviewer's constructive comments. As suggested, we have repeated our ChIP-PCR experiments and have included ChIP assays performed against unmodified histone H3, which we have used to normalize the data from the H3K9me2/3 ChIP assays.

The data is shown in revised figure 8b and Supplementary figures 15, 16 of the manuscript.

Minor points

1) Explain in the text Fig. 4b: what is DNMT1, DNMT1 M2 and DNMT1 M3.

We apologize for the confusion. We have edited the figure legend as well as the results section to clarify the labels of the different DNMT1 constructs as shown in revised Figure 5b of the manuscript.

2) On p. 5 it says “DNA methyltransferase 1 (DNMT1), a key enzyme mediating DNA methylation and histone modifications.” DNMT1 does not modify histones. There are many other misleading statements throughout.

We apologize for the confusion. The statement has been corrected to state that DNMT1 contributes to the regulation of histone modifications via altering the recruitment of chromatin remodeling enzymes.

Reviewer #2

In the current study by Yan et al. they found out that impaired wound healing in diabetic mice is mediated by an HSC-autonomous mechanism. HSCs under diabetic conditions show an increased oxidative stress response that is leading to the de-repression of the let-7 target gene DNMT1. Upregulation of DNMT1 in turn downregulates some of its target genes important for HSC differentiation towards macrophages. Finally, this dysregulation within HSCs in diabetic mice reduces macrophage infiltration and stimulates M1 polarization within the wound area. The study is interesting and the understanding of the regulation of the inflammatory response in wound healing is of high relevance. Nevertheless, there are several major technical concerns prior publication, because the manuscript is too preliminary at its present state.

Specific points:

1. It is well established that the wound healing response in db/db mice is delayed in comparison to wild type mice (Figure 1). But the finding that HSCs isolated from db/db mice delay wound healing in irradiated wild type mice and that wild type HSC rescue the wound healing response in db/db mice is essential for the current study, but only superficially analyzed. The authors should provide a detailed analysis of the wound healing response with **histological data**. The macroscopic quantification of wound closure is not sufficient to assure improved or delayed healing. This is particularly relevant since no differences can be seen between the representative images of wt->wt and db/db->wt transplanted mice (in Figure 5b) and between wt->db/db and db/db-> db/db mice (in suppl. Figure 6b). For the three studies described in Figure 2, 5 and Suppl. Figure 6 the authors should provide the following results.

Thank you for the reviewer's constructive comments. We have performed a detailed and comprehensive histological analysis of the wound healing response and we have added the results of this histological analysis as suggested.

1) quantification for the wound healing response: (1) distance between epithelial tips (2) amount of granulation tissue (3) vascularization based on histological images of wound tissue isolated at day 7, 14 and 21 post injury.

Please see revised **Figure 1c-f, Figure 3c-f, Figure 6c-e, Supplementary**

Figure 11 of the manuscript.

2) provide histological findings to illustrate the differences in macrophage infiltration and macrophage polarization.

Please see revised **Supplementary Figure 2** in which we show that following the induction of cutaneous wounds, total macrophage infiltration in *db/db* mice was significantly lower on day 7 (new tissue formation phase) than in WT mice.

3) determine vascularization of the wounds.

Please see revised **Figure 1g-i** and **Supplementary Figure 4f-h** of the manuscript.

Furthermore, the authors should add a scale bar in their images shown in Figure 1b and 5b, because it appears that the distance varies.

Thank you for the comments. When we took the macroscopic wound photographs, we used a ruler in order to avoid any biases caused by magnification, as shown below. However, in the images shown in the manuscript, we did not include the ruler in order to have a clearer view of the wounds in the different groups.

In Figure 2 the representative macroscopic images are additionally missing and should be added.

As suggested by the reviewer, we have added representative macroscopic images in revised **Figure 3b** of the manuscript.

2. The authors should add their raw data (dot plots) for the FACS analysis in Figure 1e-g, 2d-f and 5d-f including the percentage of all cell populations. Since it is not mentioned if the authors used any isotype-matched controls

for their FACS analysis, they should provide some data proving the specificity of their antibodies.

As suggested by the reviewer, schematics of the flow cytometry gating strategy, as well as an example of isotype-matched control results, were added in revised Figure 2b,e, Supplementary Figure 3a,b and Supplementary Figure 5 of the manuscript.

3. Regarding Figure 3, at no point in the manuscript it is explained what DCF+ cells are and how the experiment was done. The NAC treatment is missing in the material and method section, too. The authors should provide the missing information.

The methods section was edited to include information on the DCF experimental procedure as well as the NAC treatment. We have also added detailed results to revised Figure 4a of the manuscript.

4. It is unclear how microRNAs shown in suppl. figure 4 have been selected. Since many let-7 family members share the seed sequences, the regulation of family members should be shown in detail. Moreover, the authors should comment on the targets of let-7-5p which appears up-regulated.

The microRNAs shown in supplementary figure 4 of the original manuscript have been selected based on the fact that they are the most significantly differentially regulated in WT versus *db/db* HSC samples.

The seed sequences between let-7d-3p and let-7f-5p are different. The seed sequence for let-7d-3p is CUAUACGACCUGCUGCCUUUC and the seed sequence for let-7f-5p is UGAGGUAGUAGAUUGUAUAGUU. The let-7d-3p target sequence is present in the DNMT1 promoter as shown in revised Supplementary Figure 8b of the manuscript.

5. The differences in DNA methylation is modest (max 10 %). Is this sufficient to change gene expression? The data should be provided in more detail (% methylation of the specific cytosines at the promoters, TF binding sites should be shown).

Thank you for this thoughtful comment. We have repeated our pyrosequencing experiments and have now plotted our data to show methylation changes at individual CpG sites, as shown in revised Figure 8a and Supplementary Figure 13 of the manuscript.

We believe that, as presented, the data now shows convincingly the significant methylation changes and reinforces our claim that impaired wound healing is due to the increased methylation of Notch 1, PU.1 and KLF4.

6. What is the effect of let-7 on DNA methylation and histone modifications?

In order to address this question, we have transfected the let-7d-3p inhibitor in WT HSCs and the let-7d-3p mimic in *db/db* HSCs. We have analyzed methylation levels of Notch1, PU.1 and KLF4 using pyrosequencing, shown in **Supplementary figure 14**. While our analysis showed a significant increase in DNA methylation in Notch1, PU.1 and KLF4 in *db/db* HSCs, increased expression of let-7d-3p and knockdown of DNMT1 in *db/db* HSCs decreased the methylation of the three genes, as we had hypothesized. We have also performed ChIP-PCR analysis and looked at the levels of histone modifications in the previously cited genes. ChIP using unmodified histone H3 antibodies were used to normalize the data from the H3K9me2/3 ChIP assays. The data in **Supplementary figure 16** shows that an increased expression of let-7d-3p in *db/db* HSCs decreases the histone methylation levels in Notch1, PU.1 and KLF4.

7. The authors should provide the corresponding Western Blot analysis for the immunoprecipitation of H3K9me3, H3K9me2 and H3K9Ac.

Since DNA is crosslinked to protein during ChIP assays, one must reverse the crosslinks and digest the chromatin proteins with proteinase K in order to obtain DNA that is clean enough for qPCR. This treatment prevents analyses of the histone modifications from the same ChIP samples that are utilized for qPCR. In order to western blot the modifications enriched during ChIP, we would have to perform independent IPs in which the samples to be western blotted are not treated identically to the ChIP samples utilized for qPCR (either done in the absence of crosslinking or with more extensive heating to reverse formaldehyde crosslinks in the absence of protease). In addition, only a few % of each histone are immunoprecipitated in ChIP experiments, which would presumably be the result of performing a ChIP-western blot. In other words, it will show that a small portion of the input material is immunoprecipitated. Since this information would likely not change the conclusions of the ChIP experiment and is not routinely included in ChIP

studies, we believe that this is beyond the scope of our study.

8. The manuscript is not very well written: the introduction should end with the aim of the study, instead of recapitulating the authors own previous studies (should be included in state of the art). Also in the result section it should be better explain why the next steps were done (e.g. the regulation of redox related enzymes and next the elucidation of Dnmt1 comes without a logical link.

Thank you for the comments. We have rewritten the introduction following the reviewer's advice. We have also rewritten parts of our Results section to clarify the reasoning behind our experimental plan.

9. The literature is also not adequately covered. Previous studies report already that let-7 affects bone marrow cell functions in dbdb mice (Bae et al Arterioscler Thromb Vasc Biol. 2013 Aug;33(8):1920-7) and let-7 has additional functions in angiogenesis (e.g. Kuehbacher et al Circ Res. 2007 Jul 6;101(1):59-68).

Thank you. The two papers were added to the revised manuscript (Ref 44 and Ref 45).

Reviewer #3

This is a very interesting study demonstrating that wild type mice transplanted with HSC from T2D mice have impaired cutaneous wound healing, similar to what is observed in T2D mice. Impaired wound healing was associated with reduced monocyte/macrophage abundance in wounds, and increased M1 macrophage polarisation, as is observed in poorly healing wounds in patients with T2D. HSC from T2D mice were impaired in their differentiation along the monocyte/macrophage lineage. The mechanism underlying reduced differentiation and increased M1 polarisation was related to increased oxidative stress in T2D HSC (possibly due to hyperinsulinemia), which was found to reduce expression of Mir- let-7d-3p, which in turn increased expression of the let-7d-3p target, DNMT1. Increased DNMT1 was shown to have a causative role in the impaired wound healing, as DNMT1-deficient T2D HSC transplanted into wt mice rescued the impaired wound healing associated with T2D HSC transplant. Further, DNMT1-deficient T2D HSC could improve wound healing in T2D mice. Mechanistically, DNMT1 was implicated in repressive chromatin modification (methylation, histone acetylation) at several loci involved in macrophage differentiation and polarisation, specifically PU.1, Notch1 and KLF4. This is a novel mechanism by which T2D leads to epigenetic reprogramming of HSCs, that appears to have flow on consequences for monocyte/macrophage differentiation and wound healing – although the concept itself is not entirely novel (authors' reference 8 demonstrated a similar phenomenon with respect to wound healing in T2D).

- 1- Re monocyte/macrophage phenotyping, does the f480+/cd115+/cd11b+ phenotype include all monocytes/macrophages in bone marrow and in the wound? – Resident macrophages, in both bone marrow and peripheral tissues, are frequently cd115low/neg? This information is critical to the claims of 'reduced monocyte/macrophage differentiation' and 'infiltration' that are made throughout the manuscript. Flow gating strategies for marrow and wound populations should be shown, including M1/M2 macrophage phenotyping.

Thank you for the reviewer's thoughtful comments.

Monocytes/macrophage are heterozygous populations, and there are a variety of markers that can be used for their identification. The

f480+/cd115+/cd11b+ phenotype is based on our careful and thorough review of the literature, which showed that CD115 is one of the most stable monocyte markers for flow cytometry analysis, while cd11b is a non-specific monocyte marker (Journal of Immunological Methods, 2013(390): 1-8; J Histochem Cytochem, 2011, 59: 812; PNAS, 2011, 108: 14566-14571). That being said, we did observe similar results by quantifying F4/80+CD11b+ monocytes/macrophages in the bone marrow and the wounds. Furthermore, we have included our flow gating strategy in **Figure 2e**, and the phenotyping of M1/M2 macrophages in **Supplementary Figure 3**.

- 2- Is there a potential role for host tissue resident macrophages in healing wounds in HSC-transplanted mice? I.e cell intrinsic effects from the *db/db* HSC may influence local macrophage populations during healing?

We did not analyze the effects of transplanted HSC to local resident macrophage populations in our study since our central hypothesis is that diabetes induces stable intrinsic changes in HSCs that impairs their macrophage differentiation and polarization in cutaneous wounds. However, based on previously published work, dermal macrophages are constantly replenished from circulating monocytes (ATVB, 2015, 35: 1066; Immunity, 2013, 39:925) and the substantial contribution of BMDCs such as macrophages was demonstrated by significantly delayed wound healing (Blood. 2011 May 12;117(19):5264-72; Dis Model Mech. 2013 Nov;6(6):1434-47).

- 3- Flow cytometry data is expressed as cell proportions – absolute cell numbers would be valuable to confirm whether changes in cell proportion relates to altered cellularity in the marrow of wt v *db/db* mice. Thank you for the reviewer’s comments. We have added the absolute monocytes numbers in the bone marrow in revised **Figure 2c** of the manuscript.

E.g. in the Discussion it is claimed that “the total number of macrophages in wounds of T2D mice is significantly lower than that in WT mice”, which is not shown in the data presented.

Thank you for carefully reading our manuscript. We did not measure the absolute macrophage numbers in wounds and have corrected this error in the discussion.

Is the reduction in monocytes/macrophages in *db/db* bone marrow associated with expansion of precursor cells, or another lineage that may be detrimental to wound healing – e.g. granulocytes?

We did perform cell lineage analysis in the early stages of the project and only observed significantly decreased monocytes in *db/db* bone marrow but no effects on granulocytes as shown in revised **Supplementary Figure 1c** of the manuscript.

- 4- The hypothesis is that sustained elevated DNMT1 expression underpins the impaired differentiation of HSC along the macrophage lineage, as well as macrophage migration and maturation/polarisation in the tissue microenvironment. Is there elevated DNMT1 expression in *db/db* bone marrow or wound monocytes/macrophages; or is it just in HSC? i.e. although there are fewer of them, are the cells that are able to differentiate, able to overcome the repressive effect of DNMT1?

As requested by the reviewer, we have performed qRT-PCR and Western blot analysis from WT and *db/db* bone marrow monocytes and wound macrophages to look at DNMT1 mRNA and protein levels. As shown in revised **Supplementary Figure 7b-d**, both gene expression and western blot analyses showed that DNMT1 levels were increased in bone marrow monocytes and wound macrophages.

- 5-The data transformation and presentation as “% change” used throughout for the M1/M2 macrophage profiling (e.g. Fig 1e-f) seems to me to lose valuable information. In the current presentation, it is not possible to discern the % of monocytes/macrophages that express an M1/M2 phenotype, how this changes during wound healing in the wt setting, and how it is affected in the diabetic or manipulated setting. In addition, there no error bars on these figures, and it is not clear whether the data represent fluorescence intensity, or % positive cells, or something else? Was any co-staining for these markers performed?

Thank you for the reviewer’s insightful comments. We did have the data originally presented as % changes (**Please see **Supplementary Figure 3c-d****). These are very complex data sets with multiple groups, multiple time points and 6 different M1/M2 markers, so we believe that our data presented as a ratio of experimental versus control group would be clearer for the readers as shown in revised **Figure 2f-h, Figure 3i-k, Figure 6g-i** of the manuscript.

We added error bars in our revised figures and M1/M2 macrophages are identified by co-staining of F4/80 and representative M1 or M2 macrophage markers. Finally, we added immunohistochemical staining images in revised **Supplementary Figure 2** of the manuscript.

- 6- In Figure 4b – what are M1 and M2 – presumably mutated MiRNA binding sites, but this is not clear in methods or figure legend. Contrary to the results text only one of them affects reporter expression.

Sorry for the confusion. We have edited the original figure 4b (now **Figure 5**) legend as well as the methods and results sections to clarify the labels of the different DNMT1 constructs.

- 7- Figure 4c-f: Are some of these panels mislabeled, they do not correspond to the results text/figure legend? According to the legend, 4c-d represent DNMT1 expression in wt cells, not let7 expression as on y axis in panel c. Why would a let7 mimic decrease native miRNA expression in wt cells as shown in c? Similarly with e-f – the let7 mimic increased let7 expression, and the inhibitor decreased DNMT1 expression in *db/db* cells? There are no details of these MiRNA mimics or inhibitors in the methods section.

We apologize for the confusion and have corrected the figure legend of the original manuscript as well as the y-axes labels in order to clearly explain plots c-f. The figure is now **Figure 5** in the revised manuscript. We have also added details about the let-7d-3p mimic and inhibitor constructs in the methods section.

- 8- The selected macrophage differentiation factors PU.1, Notch1 and KLF4 play important roles in different phases of monocyte/macrophage differentiation – I was curious about the rationale for the selection of these particular genes, which all turned out to be epigenetically modified? Were other candidate genes investigated? – how specific is DNMT1 likely to be, how widespread the hypermethylation effect etc. could be discussed.

While we recognize that Notch1, PU.1 and KLF4 are not the only genes that are epigenetically modified by DNMT1 in our experimental model, these three genes were selected based on our extensive literature searches which strongly indicate the critical role of Notch 1, PU.1 and KLF4 in

monocytes/macrophage differentiation and polarization (Ref 27-36). In order to address comments made by other reviewers regarding global DNA methylation levels in *db/db* mice, in which we show a significant increase in DNMT1 protein levels (please see **Supplementary Figures 9a-b**), we have performed a 5mC analysis on genomic DNA from WT and *db/db* HSCs, shown in **Supplementary Figure 7e**. The data shows that, in *db/db* HSCs, total methylation levels are increased when compared to the WT sample.

- 9- I felt the discussion offered little more than a restatement of results. The comment in the introduction “This potentially novel mechanism may be responsible for the conflicting evidence existing between the observations made at different stages of wound healing in T2DM patients and in animal models.” should be explained/further discussed as relevant to the results of this study. Potential triggers for oxidative stress, especially NOX2, could be discussed – is there a link between hyperinsulinemia and NOX2?

Thank you for the reviewer’s helpful comments. We have significantly revised our discussion section as suggested and have added a paragraph on the link between hyperinsulinemia and Nox2 on page 18.

Furthermore, while we did not intend to imply that there was conflicting evidence between the different stages of wound healing in T2DM patients and animal models, we decided to remove this sentence, as we agree that it could be perceived as confusing.

- 10- Murine HSC cultures were “supplemented with 50ng/ml SCF, 10ng/ml TPO and 10 ng/ml Flt3 with either 6ng/ml IL-3 and 10ng/ml IL-6 or 10ng/ml M-CSF or 40ng/ml M-CSF” In what circumstances were il6/il3 used vs M-CSF and what is the relevance of this?

IL6/IL3 were added to the basic HSC culture medium while M-CSF was added to induce HSC differentiation towards monocytes/macrophages. These culture methods are based on previously published methods (PNAS, 2011, 108: 14566-14571)

It is then stated that “After 6 days, cells were induced to differentiate towards M1/M2 macrophage by changing the medium to the M1 induction medium (HSC basic medium, 5% FBS, 50ng/ml LPS and 5ng/ml IFN γ) or the M2 induction medium (HSC basic medium, 5% FBS, 10ng/ml IL-4)

overnight.” – were in vitro polarisation experiments on BM HSC performed, as this suggests?

Yes. We performed in vitro M1/M2 polarization on BM HSCs and data were shown in revised Figure 7e, and Supplementary Figure 12c of the manuscript.

11- Human HSC culture – how long was the differentiation in insulin prior to mono/mac differentiation? KLF4 expression was downregulated in response to insulin in human HSC – what happened to the expression of PU.1 and Notch1?

The differentiation in insulin prior to monocyte/macrophage differentiation was performed for 48 hours.

We have performed gene expression analysis in human HSC with and without insulin treatment using qRT-PCR for both PU.1 and Notch1. The results are presented in revised Supplementary Figure 12d of the manuscript.

REVIEWERS' COMMENTS:

Reviewer #1 (Remarks to the Author):

The authors claim that "This impaired wound healing phenotype of T2D mice is due to a Nox-2-dependent increase in HSC oxidant stress that decreased microRNA let-7d-3p, which, in turn, directly upregulated DNMT1, leading to the hypermethylation of Notch1, PU.1 and KLF4." This fairly extreme claim is not supported by the data, the manuscript is not well written (although the numerous mistakes noted by the reviewers have been largely corrected), and the experiments were not well planned. The authors fail to note the substantial literature that makes it clear that levels of DNMT1 protein are regulated almost entirely at the level of transcription. The authors could have tested db homozygous mice heterozygous for mutations in Dnmt1 (which express half the level of DNMT1) to test their hypothesis more rigorously. As it stands the hypothesis quoted above is not sufficiently supported by the data.

Reviewer #2 (Remarks to the Author):

The authors addressed most concerns raised. There are only two minor points that should be addressed:

1) Images in Figure 1i and Supp Figure 4b do not show vessel like structures. The representative images in this figure are not convincing and do not support the quantification.

The authors should provide high quality images.

2) Supp Figure 2: The authors should mark from which part of the wound they have taken the pictures. It does not look like granulation tissue since hair follicle are present and the epithelium does not look like a hyperproliferative epithelium. Comparing WT and db/db images just on the basis of these images would assume, that there is no wound healing defect at all. The authors should provide also here high quality images.

Reviewer #3 (Remarks to the Author):

The authors have satisfactorily addressed my concerns in the rebuttal and revised manuscript.

Response to Reviewers' comments:

Referee #1 comments

The authors claim that "This impaired wound healing phenotype of T2D mice is due to a Nox-2-dependent increase in HSC oxidant stress that decreased microRNA let-7d-3p, which, in turn, directly upregulated DNMT1, leading to the hypermethylation of Notch1, PU.1 and KLF4." This fairly extreme claim is not supported by the data, the manuscript is not well written (although the numerous mistakes noted by the reviewers have been largely corrected), and the experiments were not well planned. The authors fail to note the substantial literature that makes it clear that levels of DNMT1 protein are regulated almost entirely at the level of transcription. The authors could have tested db homozygous mice heterozygous for mutations in Dnmt1 (which express half the level of DNMT1) to test their hypothesis more rigorously. As it stands the hypothesis quoted above is not sufficiently supported by the data.

As we described in the manuscript, our hypothesis is that hyperinsulinemia induces Nox 2-dependent HSC oxidant stress that increases the expression of DNMT1, rather than the leptin receptor mutation. The investigation of db/db DNMT1^{-/-} or db/db DNMT^{+/-} will not necessarily produce further evidence to support our hypothesis. We did test our hypothesis by knocking down DNMT1 expression in HSCs isolated from db/db mice. In addition, db/db homozygous mice have serious reproductive defects, which will make the generation of db/db DNMT1^{-/-} and db/db DNMT1^{+/-} mice technically challenging. Therefore, we do not believe it is necessary to further test our hypothesis with db/db DNMT1^{-/-} and db/db DNMT1^{+/-} mice.

Referee #2 comments

The authors addressed most concerns raised. There are only two minor points that should be addressed:

- 1) Images in Figure 1i and Supp Figure 4b do not show vessel like structures. The representative images in this figure are not convincing and do not support the quantification. The authors should provide high quality images.

Thank you for the reviewer's comments. We appreciate the reviewer taking the time to read our revised manuscript so carefully. The suggestions have definitely helped to increase the quality of our work.

In this study, the vessels in the cutaneous wound sections were identified by immunostaining of the specific endothelial marker-CD144 and the smooth muscle cell marker— α -SMC. As the reviewer mentioned, proper morphological identification of the arteries and veins is critical for the accurate quantification of vessel density. To this end, multiple images at 400x magnification were taken in this study. Representative images from each experimental groups were chosen based on the following consideration:

1. Clear vascular structure
2. Differences in both artery and vessel density.
3. Differences among 8 different experimental groups including WT, db/db, WT HSC→WT, db/db HSC→WT, db/db HSC+shDNMT1→WT, WT HSC→db/db, db/db HSC →db/db, db/db HSC+shDNMT1→db/db.

4. Since the vessels in cutaneous wounds are not equally distributed, the representative images from different groups may not be exactly at the same spot in the wounds.

As the reviewer has suggested, we replaced some images with higher quality ones **in Figure 1 and Supp Figure 4b.**

2) Supp Figure 2: The authors should mark from which part of the wound they have taken the pictures. It does not look like granulation tissue since hair follicle are present and the epithelium does not look like a hyperproliferative epithelium. Comparing WT and db/db images just on the basis of these images would assume, that there is no wound healing defect at all. The authors should provide also here high quality images.

Thank you for the reviewer's comments. In Supp Figure 2, the representative images are mainly located at the edges of wounds (between normal skin and healing area). The representative images are chosen mainly based on the following consideration:

1. Clear cell morphology.
2. Showing double staining of general macrophage marker staining and M1 or M2 macrophage markers.
3. Difference in macrophage number and M1/M2 polarization between wild type and type 2 diabetic wounds.
4. Images are located at the edges of wounds. The difference is much clearer around this area.
5. Since the macrophages and M1/M2 phenotypes in cutaneous wounds are not equally distributed, the representative images may not be exactly at the same spot in wounds.

As the reviewer has suggested, we have added H&E staining images to mark the areas of macrophage staining **in Supplementary Figure 2.**

Referee #3 comments

The authors have satisfactorily addressed my concerns in the rebuttal and revised manuscript.

We are very thankful to the reviewer for reviewing our manuscript and making numerous insightful comments.